# GRAPH-BASED DOCUMENT STRUCTURE ANALYSIS

**Yufan Chen,   Ruiping Liu,   Junwei Zheng,   Di Wen,   Kunyu Peng,
Jiaming Zhang**[*]**   Rainer Stiefelhagen**

CV:HCI Lab, Karlsruhe Institute of Technology

{firstname.lastname}@kit.edu

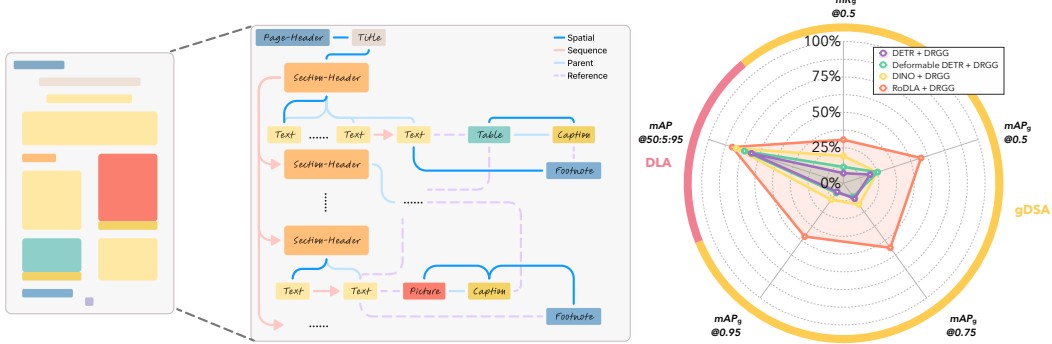

(a) Document graph structure of layouts with different relations.    (b) Performance of our proposed method.

Figure 1: **GraphDoc Dataset Overview**. Figure 1a illustrates the threefold considerations, including (i) the inclusion of spatial and logical relations, (ii) support for multiple relations between layouts pairs, (iii) and the integration of non-textual elements. Figure 1b demonstrates the state-of-the-art performance of our proposed method, showcasing mAP results for the Document Layout Analysis (DLA) task, as well as $mR_g$ and $mAP_g$ results for the graph-based Document Structure Analysis (gDSA) task on the GraphDoc dataset.

## ABSTRACT

When reading a document, glancing at the spatial layout of a document is an initial step to understand it roughly. Traditional document layout analysis (DLA) methods, however, offer only a superficial parsing of documents, focusing on basic instance detection and often failing to capture the nuanced spatial and logical relations between instances. These limitations hinder DLA-based models from achieving a gradually deeper comprehension akin to human reading. In this work, we propose a novel graph-based Document Structure Analysis (**gDSA**) task. This task requires that model not only detects document elements but also generates spatial and logical relations in form of a graph structure, allowing to understand documents in a holistic and intuitive manner. For this new task, we construct a relation graph-based document structure analysis dataset (**GraphDoc**) with 80K document images and 4.13M relation annotations, enabling training models to complete multiple tasks like reading order, hierarchical structures analysis, and complex inter-element relation inference. Furthermore, a document relation graph generator (**DRGG**) is proposed to address the gDSA task, which achieves performance with **57.6%** at $mAP_g@0.5$ for a strong benchmark baseline on this novel task and dataset. We hope this graphical representation of document structure can mark an innovative advancement in document structure analysis and understanding. The new dataset and code will be made publicly available at GraphDoc.

## 1 INTRODUCTION

Understanding the structural layout of documents is a fundamental aspect of document analysis and comprehension. Traditional Document Layout Analysis (DLA) methods primarily focus on detecting and classifying basic document elements, e.g., text blocks, images, and tables. While these

---

[*]Corresponding author.

Table 1: **Modern Document Structure Analysis Datasets**. V, T, L, O, H and G stand for Visual, Textual, Layout, Order, Hierarchy and Graph modality. DLA, ROP, HSA and GSA stand for Document Layout Analysis, Reading Order Prediction, Hierarchcial Structure Analysis and Graph Structure Analysis. NTI stands for Non-Textual Instance.

| Dataset | Year | Instance Level | Modality | | | | | | #Image | # Object Categories | # Object Instances | # Relation Categories | # Relations | NTI | Tasks | | | | Format |
| | | | T | V | L | O | H | G | | | | | | | DLA | ROP | HSA | GSA | |
|---|---|---|---|---|---|---|---|---|---|---|---|---|---|---|---|---|---|---|---|
| FUNSD | 2019 | Semantic Entity | ✓ | ✓ | ✓ | ✓ | ✗ | ✗ | 199 | 4 | 7411 | 1 | - | ✗ | ✓ | ✓ | ✗ | ✗ | Scanned |
| ReadingBank | 2021 | Word | ✓ | ✓ | ✓ | ✓ | ✗ | ✗ | 500K | - | 98.18M | 1 | - | ✗ | ✗ | ✓ | ✗ | ✗ | DocX |
| XFUND | 2022 | Semantic Entity | ✓ | ✓ | ✓ | ✓ | ✗ | ✗ | 1393 | 4 | 0.10M | 1 | - | ✗ | ✓ | ✓ | ✗ | ✗ | Scanned |
| Form-NLU | 2023 | Semantic Entity | ✓ | ✓ | ✓ | ✓ | ✗ | ✗ | 857 | 7 | 0.03M | 1 | - | ✗ | ✓ | ✓ | ✗ | ✗ | PDF |
| HRDoc | 2023 | Line | ✓ | ✓ | ✓ | ✗ | ✓ | ✗ | 66K | 14 | 1.79M | 3 | - | ✓ | ✓ | ✗ | ✓ | ✗ | PDF |
| Comp-HRDoc | 2024 | Line | ✓ | ✓ | ✓ | ✓ | ✓ | ✗ | 42K | 14 | 0.97M | 3 | - | ✓ | ✓ | ✓ | ✓ | ✗ | PDF |
| PubLayNet | 2019 | Paragraph | ✗ | ✓ | ✓ | ✗ | ✗ | ✗ | 340K | 5 | 3.31M | - | - | ✓ | ✓ | ✗ | ✗ | ✗ | PDF |
| DocLayNet | 2022 | Paragraph | ✗ | ✓ | ✓ | ✗ | ✗ | ✗ | 80K | 11 | 1.10M | - | - | ✓ | ✓ | ✗ | ✗ | ✗ | PDF |
| **GraphDoc** | 2024 | Paragraph | ✓ | ✓ | ✓ | ✓ | ✓ | ✓ | 80K | 11 | 1.10M | 8 | 4.13M | ✓ | ✓ | ✓ | ✓ | ✓ | PDF |

methods have achieved significant success in document analysis at a superficial level, they often overlook capturing the intricate spatial and logical relations that exist between different document components. This limitation hampers the ability of models to achieve a deeper, more human-like understanding of document structures. Recent advancements in deep learning and computer vision have led to the development of models that can process multimodal information, integrating visual, textual, and layout information (Gu et al., 2021; Huang et al., 2022). However, these models still lack the capability to comprehend the complex relations inherent in document layouts, particularly the spatial and logical relations that define the document structure. As shown in Table 1, existing datasets such as PubLayNet (Zhong et al., 2019) and DocLayNet (Pfitzmann et al., 2022) provide annotations for basic layout elements but do not include detailed relational information inside.

To address these challenges, we propose a novel task called graph-based Document Structure Analysis (**gDSA**), which aims to not only detect document elements but also generate spatial and logical relations in the form of a graph structure. This approach allows for a more holistic and intuitive understanding of documents, akin to how humans perceive and interpret complex layouts. For this task, we introduce the **GraphDoc** dataset, a large-scale relation graph-based document structure analysis dataset comprising 80,000 document images and over 4 million relation annotations. GraphDoc includes annotations for both spatial relations (*Up*, *Down*, *Left*, *Right*) and logical relations (*Parent*, *Child*, *Sequence*, *Reference*) between document components, e.g., *text*, *table*, and *picture*. This rich relational information enables models to perform multiple tasks like reading order prediction, hierarchical structure analysis, and complex inter-element relationship inference. To tackle the gDSA task, we propose the **Document Relation Graph Generator (DRGG)**, an end-to-end architecture designed to generate relational graphs from document layouts. DRGG combines object detection with relation prediction, capturing both spatial and logical relations between document elements. Our experiments demonstrate that DRGG achieves a mean Average Precision of 57.6% at a relation confidence threshold of 0.5 ($mAP_g$@0.5), setting a strong baseline for this novel task and dataset.

In summary, our contributions are as follows:

- We introduce **GraphDoc**, a graph-based document structure analysis dataset that provides detailed annotations of both spatial and logical relations between document components.
- We provide a comprehensive analysis of graph-based Document Structure Analysis (gDSA) paradigms and demonstrate that our **DRGG** model effectively addresses the gDSA task.
- We conduct extensive experiments on the GraphDoc dataset and upstream DLA tasks, proving the effectiveness of the gDSA approach for document layout analysis.

## 2 RELATED WORK

**Document Layout Analysis.** To analyze the document layout is a fundamental task of the document understanding. Recent advancements in deep learning (Schreiber et al., 2017; Prasad et al., 2020) treat Document Layout Analysis (DLA) as a traditional visual object detection or segmentation challenge, employing convolutional neural networks (CNNs) to address this task. Drawing inspiration from BEiT (Bao et al., 2022), compared to the CNN-based methods, DiT (Li et al., 2022) trains a document image transformer specifically for DLA, achieving promising results, albeit overlooking the textual information within documents. Beyond the single modality, UniDoc (Gu et al., 2021) and

LayoutLMv3 (Huang et al., 2022) integrate text, vision, and layout modalities within a unified architecture. Not only methods and architectures, but also benchmark datasets have achieved promising evolution. While PubLayNet (Zhong et al., 2019) and DocLayNet (Pfitzmann et al., 2022) have only two modalities, *i.e.* , visual and layout, FUNSD (Jaume et al., 2019), XFUNSD (Xu et al., 2022), ReadingBank (Wang et al., 2021) and Form-NLU (Ding et al., 2023a) have textual, visual, layout and order modalities. It is regrettable that the aforementioned datasets, despite considering other modalities, are designed solely for textual information without non-textual information consideration. HRDoc (Ma et al., 2023) and its improved version, the Comp-HRDoc dataset (Wang et al., 2024), both take into account multimodal processing of both textual and non-textual information. Additionally, they introduce a hierarchical structure as a new modality for document analysis. However, all publicly available datasets do not consider the graphical structure of document, which is crucial for both spatial and logical structure analysis of documents. In this work, we propose Graph-Doc dataset, which contains six modalities, *i.e.* , textual, visual, layout, order, hierarchy and graph, targeting complex Document Structure Analysis (DSA) tasks.

**Graphical Representation and Generation.** To construct a graph-based structured representation is a foundational step toward higher-level visual understanding. Graph-based representation Scene Graph Generation (SGG) is versatile tool for various vision-language tasks, such as image captioning (Gao et al., 2018; Yang et al., 2019b), visual question answering (Li et al., 2019; Zhang et al., 2019), content-based image retrieval (Johnson et al., 2015; Schuster et al., 2015), image generation (Johnson et al., 2018; Mittal et al., 2019), and referring expression comprehension (Yang et al., 2019a). On the other hand, in the field of natural language processing knowledge graph generation is also well-explored. Instead of building the entire global graph structures, some methods (Li et al., 2016; Yao et al., 2019; Malaviya et al., 2020) look into a simpler problem of graph completion. Alternatively, other works (Roberts et al., 2020; Jiang et al., 2020; Shin et al., 2020; Li & Liang, 2021) propose to query the pre-trained models to extract the learned factual and commonsense knowledge. CycleGT (Guo et al., 2020) is an unsupervised approach for both text-to-graph and graph-to-text generation. In this method, the graph generation process utilizes a pre-existing entity extractor, followed by a classifier for relations. Inspired by the graph generation from computer vision and natural language processing, we propose a graph-based task for document analysis called graph-based Document Structure Analysis (gDSA). gDSA refers to the task of mapping document images into a comprehensive structural graph that contains the understanding of document structure.

**Document Relation Extraction.** Document relation extraction is a crucial task in understanding the complex interactions within documents by identifying relations between document elements. ReadingBank (Wang et al., 2021) is designed for the task of reading order detection, which aims to capture the sequence of words as naturally understood by human readers. FUNSD (Jaume et al., 2019), Form-NLU (Ding et al., 2023a) and XFUND (Xu et al., 2022) focuses on extracting relations in semi-structured documents, particularly text-only forms. Addresses the challenges in scanned documents by identifying key-value pairs and relations between textual elements. PDF-VQA (Ding et al., 2023b) extends document relation extraction to multimodal documents by incorporating visual question answering techniques. This dataset requires the identification of relations between document elements within PDFs. HRDoc (Ma et al., 2023) constructs a dataset for document reconstruction but overlooks the spatial structure and the interaction between textual and non-textual elements. Our proposed GraphDoc dataset includes both spatial and logical relations between textual and non-textual elements, resulting in a comprehensive analysis of document structure.

## 3 METHODS

### 3.1 GRAPHDOC DATASET

In this section, we introduce the GraphDoc dataset, specifically developed for document layout and structure analysis. Additionally, we define the corresponding tasks and describe the annotation pipeline employed for constructing such datasets.

#### 3.1.1 TASK DEFINITION

The goals of the GraphDoc Dataset can be represented into two tasks: Document Layout Analysis (DLA) and graph-based Document Structure Analysis (gDSA). We detail definitions respectively.

**Document Layout Analysis (DLA)**. This task focuses on extracting layout information with labeled bounding box, representing layout elements within the document. For the DLA task, the setup is similar to that of the DocLayNet (Pfitzmann et al., 2022) dataset, with the layout element size being at the paragraph level except *Table* and *Picture*. The labels are categorized into 11 distinct classes: *Caption*, *Footnote*, *Formula*, *List-item*, *Page-footer*, *Page-header*, *Picture*, *Section-header*, *Table*, *Text*, and *Title*. DLA task can be represented by the following objective function:

$$\mathcal{L}_{\text{DLA}} = \sum_{i=1}^{n} \mathcal{L}_{\text{bbox}}(b_i, \hat{b}_i) + \mathcal{L}_{\text{cls}}(c_i, \hat{c}_i) \tag{1}$$

where $b_i$ and $\hat{b}_i$ are the ground truth and predicted bounding boxes for the $i$-th layout element, respectively, and $c_i$ and $\hat{c}_i$ are the corresponding class labels. The loss function $\mathcal{L}_{\text{DLA}}$ thus encapsulates both the bounding box regression loss $\mathcal{L}_{\text{bbox}}$ and the classification loss $\mathcal{L}_{\text{cls}}$.

**Graph-based Document Structure Analysis (gDSA)**. gDSA aims to extract the relational graph among layout elements within the document, which could be formed as $G = (V, E)$. For gDSA, nodes $V$ correspond to the layout elements, edges $E$ represent the relations between these layout elements, e.g., *reference*. The objective for gDSA could be expressed as:

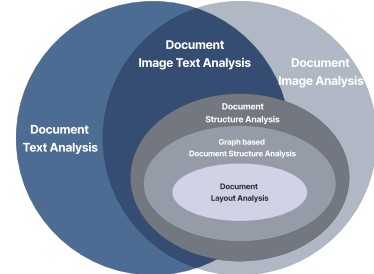

$$\mathcal{L}_{\text{gDSA}} = \sum_{(v_i, v_j) \in E} \left( \mathcal{L}_{\text{cls}}(v_i, \hat{v}_i) + \mathcal{L}_{\text{rel}}(r_{ij}, \hat{r}_{ij}) \right) \tag{2}$$

Here, $v_i$ and $\hat{v}_i$ are the ground truth and predicted labels for the layout element $i$, and $r_{ij}$ and $\hat{r}_{ij}$ represent the ground truth and predicted relations between the layout elements $i$ and $j$. The classification loss $\mathcal{L}_{\text{cls}}$ for the nodes ensures that the layout elements are accurately identified, while the relation loss

Figure 2: Overview of the GraphDoc Dataset's Task, which illustrates both DLA and gDSA tasks of GraphDoc are based on image analysis.

$\mathcal{L}_{\text{rel}}$ for the edges captures the accuracy of the predicted relations within the document's structure. Additionally, the specific functions of all the above-mentioned losses depend on the requirements of the model and the task.

Two sub-tasks can derive further from the gDSA task: Reading Order Prediction (ROP) and Hierarchical Structure Analysis (HSA). The ROP task involves determining the correct sequence in which the layout elements should be arranged. The HSA task focuses on identifying the hierarchical relations among the layout elements and establishing a structural organization within the document. In addition to the tasks described above, the gDSA task further leverages reference relations to establish connections between textual and non-textual layout elements within the document. This integration ensures that these two types of layout element are not analyzed in isolation, but rather as interconnected components. As shown in Figure 2, the gDSA tasks in the GraphDoc dataset achieves a novel and comprehensive visual analysis of document task, paving the way for novel document visual content analysis of modern complex documents.

### 3.1.2 DATASET COLLECTION

Our GraphDoc Dataset is primarily derived from the DocLayNet (Pfitzmann et al., 2022) dataset, which contains over 80,000 document page images spanning a diverse array of content types, including financial reports, user manuals, scientific papers, and legal regulations. We leveraged the existing detailed annotations and the PDF files offered through DocLayNet Dataset, to create new annotations that focus specifically on the relations between various layout elements within the documents. Additionally, in accordance with the License CDLA 1.0, users are permitted to modify and redistribute enhanced versions of datasets based on the DocLayNet dataset. Due to page limitations of DocLayNet, we will only consider relations within the same page and not those across pages.

### 3.1.3 DOCUMENT RELATIONAL GRAPHS

For visually rich documents, the spatial layout and relations between various layout elements carry significant meaning. These relations include hierarchical relations between section headers and text, sequential relations between text blocks, and references to tables or figures. Understanding these structural and relational details aids in better extraction of document information and in gaining a deeper comprehension of the document as a whole. Moreover, graphs themselves are an effective modality for enhancing the performance of scene understanding tasks.

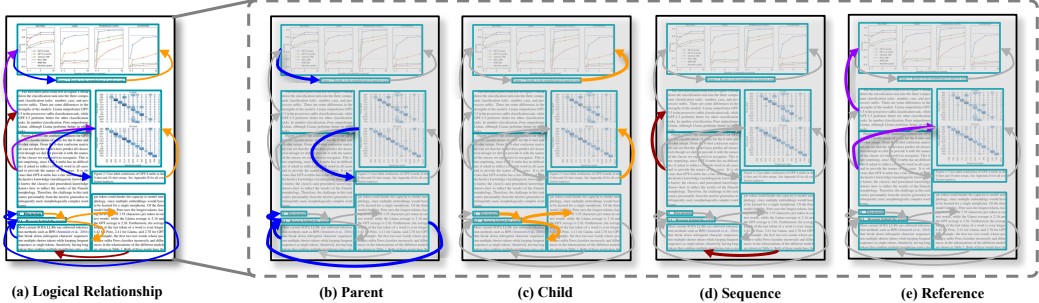

Figure 3: **Logical Relationship** in GraphDoc Dataset. There are 4 instinct types of relations. The relational graph effectively filters out extraneous connections that might appear in other types of diagrams, providing a clearer representation of the actual relationships.

Consequently, in our GraphDoc dataset, we have defined two types of relational graphs. The first type is the spatial relational graph, which primarily categorizes spatial relations into four types: *up*, *down*, *left*, and *right*. In scientific literature, the spatial structure is typically more standardized, often formatted as either two-column or single-column documents in Manhattan-Layout, which refers to a grid-like layout where content is arranged in straight, non-overlapping rectangular regions. Thus, these four spatial relations can effectively cover most of the spatial relations between layout elements within scientific documents.

The second type is the logical relation graph, which is independent of layout position and focuses on capturing the relations between layout elements from a logical structure perspective. In this logical relation graph, we categorize all relations between document layout elements into four types of relations: *parent*, *child*, *sequence*, and *reference*. All logical relations are illustrated in Figure 3 for better understanding. The detailed definitions of relation are as follows:

- **Parent**: Indicates the parent part of a parent-child. For example, a section header can be the parent of the subsection header, as in Fig. 3(b).

- **Child**: Represents the child part of a parent-child. For instance, paragraphs that belong to a section are considered children of that section header, as in Fig. 3(c).

- **Sequence**: Denotes the sequential order of layout elements. For example, the natural reading order of paragraphs in a section or the steps in a procedure, as in Fig. 3(d).

- **Reference**: Captures citation or references. For example, a figure or table being cited within the text or references to external documents, as in Fig. 3(e).

### 3.1.4 DATASET ANNOTATION PIPELINE

In order to create high-quality annotations for the GraphDoc-Dataset, we invested significant effort in enhancing the relational annotations while maintaining the foundational document layout annotations (DLA) from the original DocLayNet (Pfitzmann et al., 2022). One of the primary challenges we encountered was the complexity of accurately capturing and annotating the intricate relations between document components, particularly for tasks involving spatial and logical structures. For these challenges, we designed a heuristic rule-based relation annotation system. This system is based on the DLA task annotations and the provided PDF files from the DocLayNet dataset. The steps for relation annotating with a rule-based system are as follows:

- **Content Extraction**: We apply the Tesseract OCR [1] and PDF parser to extract the text content contained within the bounding boxes of all categories except for Table and Picture.

- **Spatial relation Extraction**: To extract spatial relations in the four directions, we heed DocLayNet annotation rules, which ensure that there is no overlap between bounding boxes. This allows us to determine spatial relations by scanning pixel by pixel along the x-axis and y-axis for spatial relations in *up*, *down*, *left*, and *right*. We record only the nearest adjacent bounding box in each direction to avoid redundant definitions.

- **Basic Reading Order**: We designed an algorithm to detect Manhattan or non-Manhattan layouts according to the spatial relation among all annotations. Additionally, we employ

---

[1] https://tesseract-ocr.github.io/

the Recursive X-Y Cut algorithm (Ha et al., 1995) to roughly establish a basic reading order based on the general left-to-right, top-to-bottom reading rule.

- **Hierarchical Structure**: Annotations were categorized into four groups based on their roles: (1) elements with direct structural relations; (2) non-textual content within the logical structure; (3) elements lacking direct associations; and (4) references. We establish an internal tree structure for the first two groups based on the text annotation, category, and basic reading order. Within the non-textual content group, *Caption* is designated as the child of the corresponding *Table* and *Picture*, to provide textual representations.

- **Relation Completion**: Using the extracted hierarchical structure, we establish *parent* and *child* relations within each group. *child* nodes under the same *parent* are sequentially ordered via *sequence* relations based on basic reading order. We match annotation texts to construct *reference* relation. The *reference* relations among *Table* and *Picture* are established, excluding *Caption*. However, *references* within *Caption* to others are maintained.

In summary, we developed a rule-based relation annotation system that efficiently constructs instance-level relational graph annotations, aligned with document elements bounding box and category annotations for the gDSA task. Moreover, the most of the results have been manually verified and refined. Our annotation system captures the inherent spatial and logical relations of document layouts, resulting a robust foundation for training and evaluating models on complex DSA tasks.

### 3.1.5 DATASET STATISTICS

In total, the GraphDoc dataset extends DocLayNet (Pfitzmann et al., 2022) by enriching it with detailed relational annotations while maintaining consistency in instance categories and bounding boxes. It comprises $80,000$ single-page document images, each selected from an individual document, resulting in $1.10$ million instances across 11 categories: *Caption*, *Footnote*, *Formula*, *List-item*, *Page-footer*, *Page-header*, *Picture*, *Section-header*, *Table*, *Text*, and *Title*. We have expanded the relational data into eight categories as defined in Sec. 3.1.3, yielding $4.13$ million relation pairs. Spatial relations constitute $64.06\%$ of these pairs, while logical relations make up the remaining $36.94\%$. It shows that spatial relations dominate the dataset, reflecting the structured nature of document layouts, where components such as *Section-header*, *Page-footer*, and *Text* are frequently positioned in spatial proximity. Logical relations, although comprising a smaller portion, play a critical role in linking elements, e.g., *Table* and *Picture* to the corresponding *Text*.

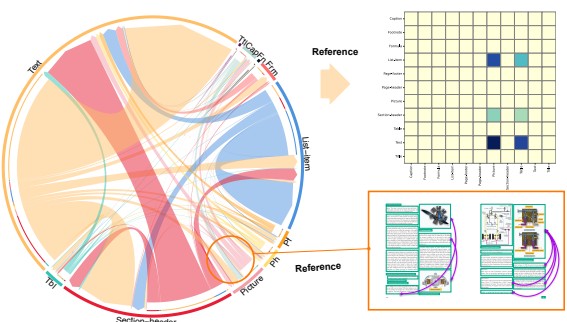

The detailed distribution of these relation pairs is illustrated in Figure 4, which provides a comprehensive overview of the relational statistics within the dataset. The left side of Figure 4 presents an aggregate view of the total relation flow between different object categories, disregarding the specific types of relations (e.g., spatial or logical). This visualization highlights how various document elements, such as *Text*, *Picture*, and *Section-header*, interact within the dataset. The intensity of relation flow between categories such as *Text* and *Picture* underscores the typical structure of documents, where these elements frequently co-occur or are positioned in proximity to one another.

Figure 4: Relation statistics on the GraphDoc dataset. The chord diagram on the left illustrates the distribution of relationships among various layouts. The heatmap on the right visualizes the intensity of relations based on layouts (deeper color means higher intensity). Below the heatmap, a detailed image presents the case of **Reference** relations for **Picture**.

On the right-hand side of Figure 4, the figure delves deeper into a specific relation type *reference*. The top section presents a heatmap that captures the frequency and distribution of reference relations between different object categories. This heatmap highlights that category *Table* and *Picture* have significantly intensive interactions observed with other document layout elements, e.g., *Text* and *List-item*. The lower section provides concrete examples of these reference relations, illustrating the detailed reference situation of **Picture** in a real-world document context. Together, these visualizations offer a holistic view of both the overall relational patterns and the specific behaviors of *reference* relations, providing deeper insights into the structural complexity of document layouts.

## 3.2 Document Relation Graph Generator

In this section, we introduce the Document Relation Graph Generator (DRGG), an architecture designed to generate instance-level relational graphs. DRGG provides an end-to-end solution to construct graphs that capture both spatial and logical relations between document layout elements. By leveraging visual features, DRGG aims to detect and analyze the structure of document layout elements accurately.

As depicted in Figure 5, the proposed model is based on Encoder-Decoder architecture with backbone for feature extraction. The backbone extracts low-level features from the document image, which are refined through the Encoder-Decoder framework. These refined features are processed through two main heads: the object detection head, responsible for document layout analysis task, and the relation head (DRGG), which predicts relations. DRGG is designed as a plug-and-play component, enabling seamless integration with existing models without requiring any modification. DRGG consists of two parts: relation feature extractor and relation feature aggregation.

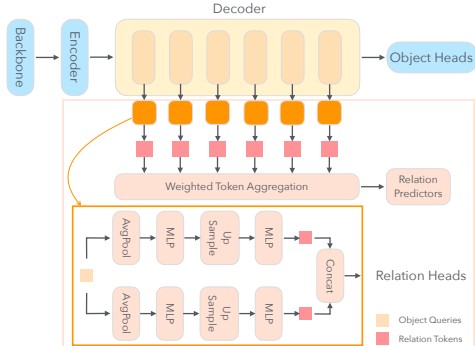

Figure 5: Proposed Document Relation Graph Generator (DRGG) for Document Layout Analysis and Document Structure Analysis. The key of our model is illustrated in the Relation Head, which is responsible for predicting relations between layout elements. The remaining parts are the standard encoder-decoder architecture used for object detection.

**Relation Feature Extractor**. The object queries $(X^0)$ and object feature representations $(X^l)$ calculated at each decoder layer $l$ are fed into independent relation feature extractors in DRGG respectively. These are then processed separately through two independent pooling layers $(P)$ and Multi-Layer Perceptrons $(\text{MLP}_p)$ in extractors as follows:

$$D_1^l = \text{MLP}_p^1(P_1(X^l)), \quad D_2^l = \text{MLP}_p^2(P_2(X^l)), \tag{3}$$

where $X^l \in \mathbb{R}^{N \times d_{\text{embed}}}, D_{1,2}^l \in \mathbb{R}^{N \times d_{\text{pool}}}$. Pooling aggregates information across channels, reducing redundancy and improving robustness. The extracted one-dimensional relational features are then through upsampling layer $(U)$ and further refined through MLP layers $(\text{MLP}_u)$, then concatenated with the original object features to form a unified representation of the relational feature $(D)$. The two representations are subsequently expanded into two dimensions along different axes and concatenated to derive the final relational features:

$$F^l = \text{Concat}(\sigma(\text{MLP}_u^1(U_1(D_1^l)) + X^l) \otimes \mathbf{1}_{d_{\text{embed}}}, \sigma(\text{MLP}_u^1(U_1(D_2^l)) + X^l)^T \otimes \mathbf{1}_{d_{\text{embed}}}), \tag{4}$$

where $F^l \in \mathbb{R}^{N \times N \times 2d_{\text{embed}}}$. This approach captures both direct relations, e.g., spatial proximity, and indirect relations, e.g., *reference*, between elements.

**Relational Feature Aggregation**. The extracted relation features from each decoder layer are combined using a weighted aggregation method to form a unified representation of the relations between all object queries. This unified representation is subsequently incorporated into the relation predictor $(\text{MLP}_g)$ to generate the relational graph prediction:

$$G = MLP_g\left(\sum_{l=1}^{L} \alpha^{(l)} F^l\right), \tag{5}$$

where $G \in \mathbb{R}^{N \times N \times k}$, $k$ is number of relation category. $\alpha^{(l)}$ are learnable weights for token aggregation. This query-based mechanism ensures that the final document relation graph, which represents a combination of image features, spatial layout, and semantic relations, would also be able to improve the accuracy of both document layout analysis and relational prediction.

The output of DRGG is a well-structured graph where nodes represent document elements, and edges represent the relations between these elements. By combining DLA result from detection head, DRGG ensures a more detailed and accurate representation for document structure analysis. More details about the DRGG architecture are presented in the supplementary Sec. C.

## 3.3 Evaluation Metrics for gDSA

In traditional Scene Graph Generation (SGG) evaluations, metrics such as Mean Recall@$k$ and Pair-Recall@$k$ assess the top-$k$ subject-predicate-object triplets ranked by predicted confidence

scores (Lorenz et al., 2024). However documents often contain a variable number of relations, and limiting the evaluation to a fixed top-$k$ can result in important relations being overlooked if they are not among the top predictions. Furthermore, there is a significant class imbalance in the relations within documents: spatial relations are prevalent, whereas logical relations such as *reference* are relatively rare. This imbalance poses challenges for evaluation metrics that rely on top-$k$ filtering. Threshold-based filtering, in contrast, allows for the inclusion of all relations that exceed a certain threshold, regardless of their frequency or ranking. This approach ensures that rare but critical relations are adequately considered during evaluation. Moreover, unlike in traditional SGG, where typically only one relation exists between subject-object pairs, layout elements in the gDSA task can have multiple coexisting relations (e.g., spatial and logical relations), both of which are essential for understanding the document structure. Therefore, the proposed evaluation metrics, $mR_g$ and $mAP_g$, should be capable of measuring the performance in both aspects: detecting layout elements and identifying multiple relations between them, including less frequent but significant relations.

To address these challenges, we first perform an exact matching of predicted instances to ground-truth instances based on both bounding box overlap and object category correspondence. Once this mapping is established, we evaluate the predicted relations within this matched set. Similar to the Intersection over Union (IoU) threshold used in object detection, we introduce a relation confidence threshold $T_R$. All relations with confidence scores exceeding this threshold are considered positive relation predictions. The remaining settings align with standard SGG evaluation metrics. This method ensures that the relation evaluation depends on both the performance of document layout analysis and the relation predictions. By explicitly considering the impact of bounding box detection and label prediction on the quality of relation predictions, our evaluation provides a comprehensive assessment of the gDSA task. The detailed algorithmic process is presented in Algorithm 1.

---

**Algorithm 1** Relation Graph Evaluation Metrics $mR_g@T_R$ and $mAP_g@T_R$ for gDSA Task

---

**Input:** Predicted instances $I_{out}$, ground truth instances $I_{gt}$, predicted relations $R$, ground truth relations $R_{gt}$, IoU threshold $T_{IoU}$, relation score threshold $T_R$
**Output:** Mean Recall $mR_g@T_R$, Mean Average Precision $mAP_g@T_R$
1: **Step 1: Instance Matching**
2: Initialize mapping $M[x] = $ null for each $x \in I_{gt}$
3: **for all** $i \in I_{out}$ **do**
4:     Find $x \leftarrow \arg\max_{g \in I_{gt},\, \text{label}(g)=\text{label}(i)} \text{IoU}(g, i)$
5:     **if** $x \neq$ null and $\text{IoU}(x, i) > T_{IoU}$ and $(M[x] = $ null or $\text{IoU}(x, i) > \text{IoU}(x, M[x]))$ **then**
6:         $M[x] \leftarrow i$
7: $L \leftarrow$ inverse mapping of $M$
8: **Step 2: Relation Evaluation**
9: $G \leftarrow \{(x_s, p, x_o) \in R_{gt} \mid x_s, x_o \in I_{gt}\}$
10: $X_T \leftarrow \{(i_s, p, i_o, s_p) \in R \mid s_p > T_R \text{ and } L(i_s), L(i_o) \neq \text{null}\}$
11: $mR_g@T_R \leftarrow f_{mR}(X_T, G)$         ▷ Calculate mean recall at threshold $T_R$
12: $mAP_g@T_R \leftarrow f_{mAP}(X_T, G)$     ▷ Calculate mean average precision at threshold $T_R$
13: **return** $mR_g@T_R, mAP_g@T_R$

---

## 4 EXPERIMENTS

### 4.1 COMPARED METHODS

To evaluate the effectiveness of our proposed DRGG framework on the GraphDoc dataset, we conducted experiments comparing it with several state-of-the-art methods in document layout analysis (DLA) and graphical structure analysis (GSA), including **DETR** (Carion et al., 2020), **Deformable DETR** (Zhu et al., 2021), **DINO** (Zhang et al., 2022), and **RoDLA** (Chen et al., 2024). These methods represent a broad range of approaches in object detection and relation extraction. We further explore the impact of various backbone architectures, including **InternImage** (Wang et al., 2023), **ResNet** (He et al., 2016), **ResNeXt** (Xie et al., 2017), and **Swin Transformer** (Liu et al., 2021), across these models. This allows us to understand the influence of different combination of feature extraction backbones and detector on the overall performance of the models.

### 4.2 IMPLEMENTATION DETAILS

For a fair comparison, we train and evaluate all methods in the MMDetection (Chen et al., 2019) framework. All experiments were conducted using the GraphDoc dataset for both training and

validation. To evaluate the performance of our proposed end-to-end model, we jointly trained and evaluated the DLA and gDSA tasks without separating them. For the object detector component, we employed the model's original configuration. More details are in Appendix F.

## 4.3 EVALUATION METRICS

To assess the performance of the models on the DLA and gDSA tasks, we employ a set of evaluation metrics tailored to capture both the layout elements' detection accuracy and the correctness of the predicted relations. For the DLA task, we utilize the mean Average Precision (mAP) at multiple Intersections over Union (IoU) thresholds,i.e., mAP@50:5:95. This metric computes the average precision across IoU thresholds ranging from 0.50 to 0.95 in increments of 0.05. It accounts for both the localization accuracy of the bounding boxes and the classification accuracy of the layout element categories. In the gDSA task, we report $mR_g$ at confidence thresholds of 0.5 and $mAP_g$ at confidence thresholds of 0.5, 0.75, and 0.95. By employing these metrics, we ensure a comprehensive evaluation of both the detection of document layout elements and their complex relational structures, reflecting the real-world challenges of document structure analysis tasks.

## 4.4 RESULTS

In this section, we evaluate our proposed DRGG with several models on the GraphDoc dataset to benchmark DLA and gDSA tasks. More detailed results of DRGG design are in Appendix E.

**Document Layout Analysis.** Table 2 presents the results of the DLA task, where we report the mean Average Precision (mAP@50:5:95) for different combinations of backbones and object detectors. Our proposed DRGG framework, integrated with the InternImage backbone and the RoDLA detector, achieves mAP of 81.5%, surpassing all other combinations, including the original setup without DRGG. This result highlights the effectiveness of integrating a powerful backbone with a detector specifically optimized for document layout analysis. Among the other detectors evaluated, DINO achieves mAP of 79.5% with the InternImage backbone, showing competitive performance. Deformable DETR and DETR obtain lower mAP scores of 73.4% and 68.2%, respectively, indicating challenges in capturing complex document layouts with these models. When analyzing the impact of different backbone networks using the RoDLA in combination with DRGG, the InternImage backbone consistently outperforms others. Specifically, InternImage achieves mAP of 81.5%, compared to 77.9% with ResNeXt, 73.7% with Swin Transformer, and 71.0% with ResNet. These results suggest that the advanced feature extraction capabilities of InternImage are crucial for accurately detecting and classifying diverse layout elements in complex documents.

Table 2: DLA and gDSA Task Results with DRGG on GraphDoc Dataset. **mAP@50:5:95** denotes the mean Average Precision(mAP) computed at IoU thresholds ranging from 0.50 to 0.95 in increments of 0.05 in DLA Task. **$mR_g$@0.5** denotes the mean Recall(mR) in gDSA Task for relation confidence threshold 0.5. **$mAP_g$@0.5**, **$mAP_g$@0.75**, and **$mAP_g$@0.95** denote the mean Average Precision in gDSA Task for relation confidence threshold 0.5, 0.75, and 0.95, respectively.

| Backbone | Detector | Relation Head | DLA | gDSA | | | |
|---|---|---|---|---|---|---|---|
| | | | mAP@50:5:95 | $mR_g$@0.5 | $mAP_g$@0.5 | $mAP_g$@0.75 | $mAP_g$@0.95 |
| InternImage | RoDLA | - | 80.5 | - | - | - | - |
| InternImage | DETR | **DRGG (Ours)** | 68.2 | 7.1 | 19.8 | 13.5 | 7.5 |
| | Deformable DETR | | 73.4 | 11.5 | 25.4 | 11.8 | 8.5 |
| | DINO | | 79.5 | 19.2 | 25.2 | 18.7 | 14.5 |
| | RoDLA | | **81.5** | **30.7** | **57.6** | **56.3** | **46.5** |
| ResNet | RoDLA | **DRGG (Ours)** | 71.0 | 13.8 | 45.8 | 17.6 | 13.3 |
| ResNeXt | | | 77.9 | 16.9 | 40.3 | 18.4 | 13.6 |
| Swin | | | 73.7 | 11.4 | 26.1 | 13.5 | 7.9 |
| InternImage | | | **81.5** | **30.7** | **57.6** | **56.3** | **46.5** |

**Graph based Document Structure Analysis.** For the gDSA task, we evaluate the models using mean Recall ($mR_g$@0.5) and mean Average Precision at different relation confidence thresholds ($mAP_g$@0.5, $mAP_g$@0.75, and $mAP_g$@0.95). As shown in Table 2, the combination of InternImage, RoDLA and DRGG achieves superior performance across all metrics. Specifically, it attains a mean recall of 30.7% and the highest mean average precision scores of **57.6%** at a 0.5 threshold,

56.3% at 0.75, and 46.5% at 0.95. Comparatively, other models exhibit significantly lower performance on the gDSA task. DINO, despite performing well on the DLA task, achieves a mean recall of 19.2% and a mean average precision of 25.2% at a 0.5 threshold. Deformable DETR and DETR perform even worse, with mean recalls of 11.5% and 7.1%, respectively. These results emphasize the difficulty of accurately predicting relational structures in documents and demonstrate the effectiveness of our proposed DRGG framework in addressing this challenge. Examining different backbones with the RoDLA and DRGG further highlights the importance of the backbone network in gDSA performance. The InternImage backbone consistently yields the best results, with significant margins over ResNeXt, Swin Transformer, and ResNet. This suggests that capturing complex relational information in documents requires not only specialized detectors but also powerful feature extraction capabilities provided by advanced backbone networks.

**Relation prediction analysis per category** To gain deeper insights into the model's performance on different types of relations, we present per-category relation detection results in Table 3. Our DRGG model with InternImage and RoDLA achieves the highest Average Precision ($AP_g@0.5$) across almost all relation categories. For spatial relations, *left* and *right*, the model achieves near-perfect scores of 99.0%, indicating exceptional ability to capture spatial positioning between layout elements. In *up* and *down* relations, it attains impressive scores of 49.0% each, outperforming other models by substantial margins. In logical relations, *parent* and *child*, the model achieves scores of 45.5% for both, demonstrating effectiveness in identifying hierarchical structures within documents. For the *sequence* relation, critical for understanding reading order, the model attains an AP of 56.4%, significantly higher than other configurations. The *reference* relation remains challenging, with the highest AP being 18.8% achieved by ResNeXt with RoDLA. Our model achieves an AP of 16.8% in this category. The lower performance in *reference* relations suggests that further work is needed to improve the detection of less frequent and more complex relations, possibly by incorporating textual content understanding or additional context.

Table 3: Per-category relation detection results with DRGG model on the GraphDoc dataset, evaluated with AP at relation confidence threshold of 0.5 ($AP_g@0.5$).

| Backbone | Detector | Relation Head | Up | Down | Left | Right | Parent | Child | Sequence | Reference |
|---|---|---|---|---|---|---|---|---|---|---|
| InternImage | DETR | DRGG (Ours) | 32.4 | 29.7 | 8.9 | 8.9 | 22.8 | 18.8 | 27.7 | 8.9 |
| | Deformable DETR | | 16.8 | 19.8 | **99.0** | 11.9 | 12.9 | 12.9 | 20.8 | 8.9 |
| | DINO | | 37.1 | 38.3 | 18.8 | 18.8 | 11.9 | 15.8 | 53.5 | 7.6 |
| | RoDLA | | **49.0** | **49.0** | **99.0** | **99.0** | **45.5** | **45.5** | **56.4** | **16.8** |
| ResNet | RoDLA | DRGG (Ours) | 15.1 | 17.2 | 27.7 | 27.7 | 6.9 | 4.0 | 17.8 | 16.8 |
| ResNeXt | | | 23.6 | 24.6 | **99.1** | **99.1** | 11.9 | 11.9 | 33.7 | **18.8** |
| Swin | | | 18.8 | 19.8 | 33.7 | 99.0 | 3.9 | 3.8 | 23.5 | 5.6 |
| InternImage | | | **49.0** | **49.0** | 99.0 | 99.0 | **45.5** | **45.5** | **56.4** | 16.8 |

## 5 CONCLUTION

In this paper, we introduced the GraphDoc dataset and proposed a novel graph-based document structure analysis (gDSA) task. By capturing spatial and logical relations among document layouts, we significantly enhanced the understanding of document structures beyond traditional layout analysis methods. Furthermore, we developed the DRGG, an end-to-end architecture that effectively generated relational graphs reflecting the complex interplay of document layouts. As an auxiliary module, DRGG leveraged both spatial and logical relations to improve document structure analysis tasks. We conducted extensive experiments, and the results demonstrated that DRGG achieved superior performance on the gDSA task, attaining an $mR_g@0.5$ of 30.7% and $mAP_g@0.5$, 0.75, and 0.95 scores of 57.6%, 56.3%, and 46.5%, respectively. This performance enhanced the effectiveness of combining document layout analysis with relation prediction to capture document structures.

**Limitations**. Our model structure focused only on visual modality input without multi-modality input consideration, which may have influenced the performance of complex document structure analysis. Future work should explore this integration to enhance the model's performance on relational graph prediction. Additionally, our dataset and approach were primarily designed for single-page documents, and extending them to effectively include multi-page documents posed a challenge that remained unaddressed. We acknowledged these limitations and believed that addressing them would be essential for making significant strides toward achieving a human-like understanding of documents, paving the way for intelligent document processing systems.

## REPRODUCIBILITY STATEMENT

In this section, we outline the efforts made to ensure the reproducibility of our work. All essential details necessary for reproducing our dataset, model, evaluation metrics, and results can be found in the main paper and the appendix. The data annotation process, including how we prepared the relation annotations, is detailed in Section 3.1.4 and Appendix A.1. The model architecture and implementation specifics, including hyperparameters and training configurations, are described thoroughly in Section 3.2 and 4.2 and detailed in Appendix C and F. Lastly, the calculations for the evaluation metrics, including all necessary references to ensure exact reproduction, are documented in Section 3.3, 4.3 and Appendix B.

## ACKNOWLEDGMENTS

This work was supported in part by Helmholtz Association of German Research Centers, in part by the Ministry of Science, Research and the Arts of Baden-Württemberg (MWK) through the Cooperative Graduate School Accessibility through AI-based Assistive Technology (KATE) under Grant BW6-03, and in part by Karlsruhe House of Young Scientists (KHYS). This work was partially performed on the HoreKa supercomputer funded by the MWK and by the Federal Ministry of Education and Research, partially on the HAICORE@KIT partition supported by the Helmholtz Association Initiative and Networking Fund, and partially on bwForCluster Helix supported by the state of Baden-Württemberg through bwHPC and the German Research Foundation (DFG) through grant INST 35/1597-1 FUGG.

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

# A    DETAILS OF GRAPHDOC DATASET

## A.1    RULE-BASED RELATION ANNOTATION SYSTEM

In this subsection, we provide an in-depth explanation of the methodologies employed in our rule-based relation extraction system. This detailed account covers the technical aspects of each step, which were briefly outlined in the main text.

**Content Extraction**: We extract textual content from all bounding boxes except those labeled as *Table* and *Picture* by combining Optical Character Recognition (OCR) and direct text extraction from PDF files. Initially, we utilize pdfplumber [2] to extract text and positional information directly from PDFs, enabling accurate mapping of text snippets to their corresponding bounding boxes. For regions where direct extraction is ineffective—such as scanned documents or encrypted PDFs—we apply Tesseract OCR [3] configured with appropriate language settings. By selectively employing OCR only when necessary, we enhance both the efficiency and accuracy of the content extraction process. Integrating both methods ensures comprehensive and reliable retrieval of textual information across various document types and qualities.

**Spatial Relation Extraction**: To determine spatial relations in the four cardinal directions, we leverage the non-overlapping property of bounding boxes ensured by the DocLayNet annotation rules. For each bounding box, we calculate its center point and identify the nearest neighboring bounding box in each direction by checking for horizontal and vertical overlaps. If two bounding boxes overlap horizontally, we consider them for *left* or *right* relations; if they overlap vertically, we consider them for *up* or *down* relations. We compute edge distances only when the bounding boxes do not overlap in the respective direction, ensuring accurate neighbor identification. Recording only the nearest neighbor in each direction maintains simplicity and avoids redundancy. This approach efficiently constructs a spatial map of document elements, which is crucial for understanding the layout and for subsequent processes like determining the reading order and building hierarchical structures.

**Basic Reading Order**: We establish a basic reading order that mirrors natural human reading patterns. First, we analyze the document layout to determine if it follows a Manhattan (grid-like) or non-Manhattan structure by assessing alignment consistency and spacing uniformity. We then apply the Recursive X-Y Cut algorithm (Ha et al., 1995) to segment the page hierarchically based on whitespace gaps. This algorithm recursively divides the page into smaller regions, creating a tree structure where leaf nodes correspond to individual bounding boxes. We traverse this tree in a depth-first manner, ordering the content from left to right and top to bottom, adjusted for the document's language and layout specifics. For multi-column layouts, we modify the traversal to process content column by column, respecting the intended flow. This method provides a logical reading sequence that aligns with human expectations and supports tasks like text extraction and summarization.

**Hierarchical Structure**: We organize the document elements into a hierarchical structure that reflects their logical relations. Annotations are grouped into four categories:

- Elements with direct structural relations (*Section-Header*, *Text*, *Formula*, *List-Item*);
- Non-textual content within the logical structure (*Table*, *Picture*, *Caption*);
- Elements lacking direct associations (*Page-Header*, *Page-Footer*, *Title*);
- References only (*Footnotes*)

For the first group, we construct the hierarchy by linking each *Section-Header* to the subsequent content elements (*Text*, *Formula*, *List-Item*) that belong to that section, based on the established reading order. Subsections are nested under their respective higher-level sections, creating a tree structure that mirrors the document's outline. For the second group, we associate each *Caption* with its corresponding *Table* or *Picture* based on their proximity in the document. The combined *Table/Picture* and *Caption* units are then placed into the hierarchy at positions determined by the reading order, linking them to the relevant sections or subsections. This hierarchical arrangement effectively captures the logical structure of the document, facilitating tasks such as information retrieval and semantic analysis by reflecting the inherent relations among the document elements.

---

[2]https://github.com/jsvine/pdfplumber
[3]https://tesseract-ocr.github.io/

**Relation Completion**: Building on the hierarchical structure, we establish *Parent*, *Child*, *Sequence*, and *Reference* relations among the elements. Child nodes under the same parent are connected via sequence relations that reflect the established reading order, with attributes indicating their positional sequence. Reference relations are identified by scanning the text for markers such as citations and footnote indicators, linking them to corresponding elements:

- *Footnotes*: Superscript numbers or symbols in the text are linked to *Footnote* elements.
- *Tables and Figures*: Mentions, for example, 'see Table 1' are linked to the respective *Table* or *Picture* elements.

We exclude *Caption* elements from being directly referenced to avoid redundancy but maintain references within captions to other elements. Consistency and integrity checks are performed to ensure all relations are correctly established, resolving any conflicts based on predefined rules.

While documents from various domains may have unique characteristics, adopting a consistent and general rule of relations allows for a unified approach to structure analysis. To address domain-specific nuances and ensure accuracy, we incorporate human verification, which helps adapt our method to diverse document domains while maintaining relation type definition principles. The extensive human verification and refinement cover approximately 58.5% of the dataset. We reviewed 4,852 pages of Government Tenders, 12,000 pages of Financial Reports, 6,469 pages of Patents, and 8,000 pages from other domains. The refinement rates for relation labels varied across domains: approximately 23% for Financial Reports, 8% for Scientific Articles, 26% for Government Tenders, and 17% for Patents. Based on our comprehensive cross-validation evaluation, we believe that our dataset is high-quality for the proposed gDSA task. We hope our new dataset and benchmark can provide an innovative advancement in DSA and document an understanding research field.

## A.2 DETAILED STATISTICS OF GRAPHDOC DATASET

In this section, we provide detailed statistics on the GraphDoc dataset. Building upon Do-cLayNet (Pfitzmann et al., 2022), GraphDoc extends it with rich relational annotations while maintaining coherence in instance categories and bounding boxes for a comprehensive analysis of document structures. As shown in Figure 6i, spatial relations constitute a significant portion of the relational data, representing more than half of all annotated relations. Of the remaining logical relations, *parent* and *child* and *sequence* relations dominate, while *reference* relations form a comparatively smaller subset. This distribution highlights the relation dataset appears to be imbalanced, which could easily lead to long-tail problems during model training.

**Spatial Relations** in the dataset are dominated by four types: *down*, *up*, *left*, and *right*, each representing the relative positioning of document components. Spatial relations are essential in document structure analysis because they provide contextual information beyond the raw bounding boxes of document layout elements. Simply knowing the positions of elements is insufficient for understanding the document's relational structure, especially when real-world perturbations occur, e.g., document image rotation and translation. By defining four fundamental spatial relation types, we aim to capture how document elements interact within a document fundamentally, facilitating a more robust and generalized understanding across different domains. As represented in Figures 6a and 6b, document elements *Section-header* and *Text* commonly follow a vertical arrangement, positioned above *Text*, reflecting a conventional reading order. This vertical structuring is consistent across most document types and contributes to an intuitive user experience when processing document layouts. In addition, as illustrated in Figures 6c and 6d, *left* and *right* relations account for another significant portion of spatial proximity relations. Understanding these left-right positional relations is critical when reconstructing the visual layout during document parsing tasks, as they often indicate the intended grouping of related elements.

**Logical Relations** are essential for understanding both the hierarchical and contextual connections between document layouts. These include *parent*, *child*, *sequence*, and *reference* relations, each contributing to the logical structure within documents. *Parent* and *child* relations define the hierarchical structure of document elements.As observed in Figures 6e and 6f, logical relations provide a clearer horizon compared to spatial relations. *Captions* are primarily the children of *Picture* and *Table*, while *Section-header* often serves as the parent of *Text*, *Formula*, and *List-item*. These relations are fundamental to defining the document's logical structure, as they guide the flow of information

and the progression from one element to another. Additionally, *sequence* relations are important for capturing the order in which document components should be read or interpreted. Figure 6g indicates that *sequence* relations mainly occur among *Text*, *List-item*, and *Formula* categories. Figure 6h demonstrates that *reference* relations, while limited in number, are critical for linking different parts of the document. These relations typically appear among *List-item*, *Text*, *Table*, and *Picture* elements, forming cross-references that provide additional context or clarification. While reference relations constitute a smaller fraction of the overall relational data, their significance cannot be overlooked, as they are key to understanding interdependencies between document elements.

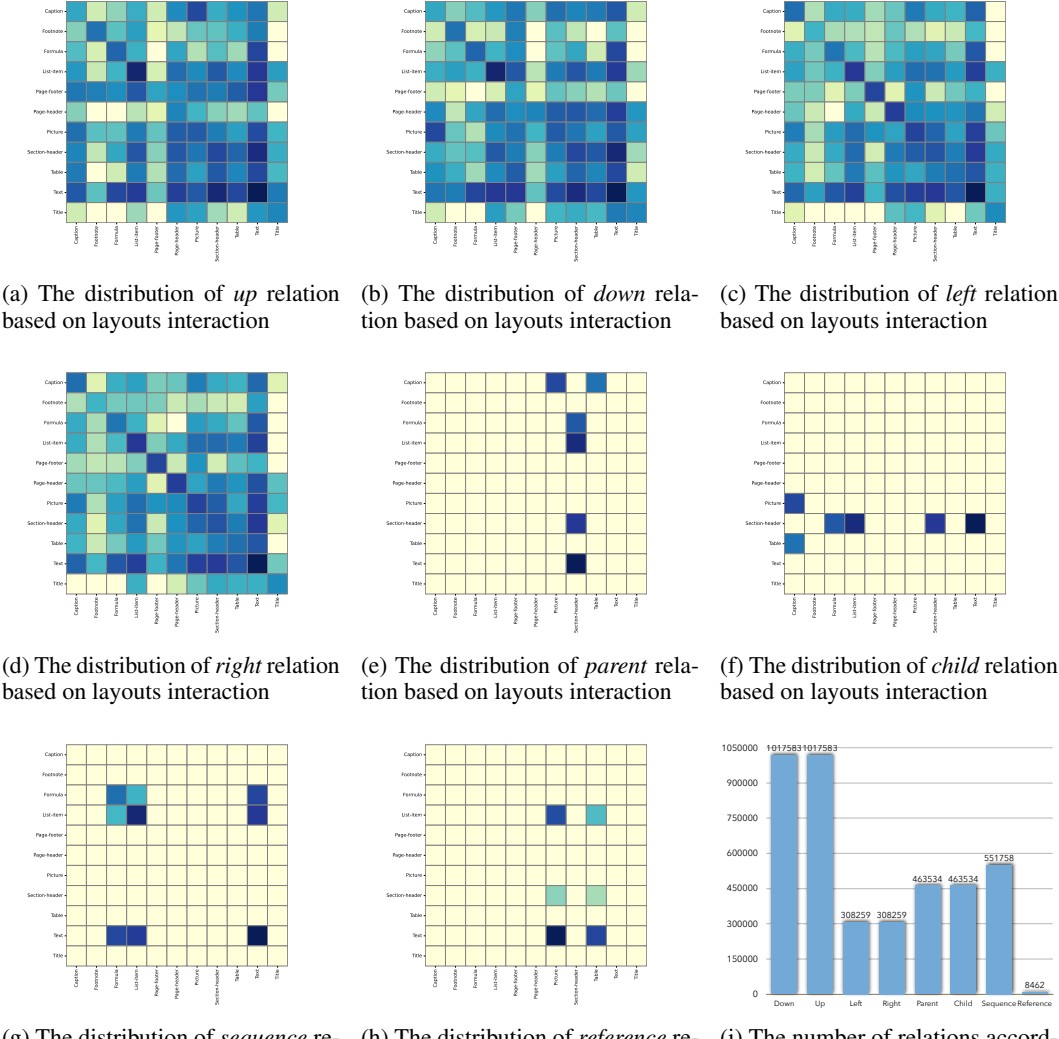

(a) The distribution of *up* relation based on layouts interaction

(b) The distribution of *down* relation based on layouts interaction

(c) The distribution of *left* relation based on layouts interaction

(d) The distribution of *right* relation based on layouts interaction

(e) The distribution of *parent* relation based on layouts interaction

(f) The distribution of *child* relation based on layouts interaction

(g) The distribution of *sequence* relation based on layouts interaction

(h) The distribution of *reference* relation based on layouts interaction

(i) The number of relations according to relation type.

Figure 6: The overview of relation distribution on GraphDoc Dataset.

# B    EVALUATION METRICS

This section details the evaluation metrics for assessing model performance on the document layout analysis (DLA) and graph-based document structure analysis (gDSA) tasks. Specifically, we discuss the Mean Average Precision (mAP) for the DLA task, and the Mean Recall (mR) and Mean Average Precision for relations ($mAP_g$) in the gDSA task.

**Mean Average Precision for DLA (mAP)**. For the DLA task, we employ the mAP over multiple Intersection over Union (IoU) thresholds, denoted as mAP@[50:5:95]. To compute the mAP, we first

calculate the Average Precision (AP) for each class $c$ at each IoU threshold $t \in 0.50, 0.55, \ldots, 0.95$ by integrating the area under the precision-recall curve. Then, we average the APs over all classes and IoU thresholds:

$$\text{mAP} = \frac{1}{|T|} \sum_{t \in T} \left( \frac{1}{C} \sum_{c=1}^{C} \text{AP}_c(t) \right), \tag{6}$$

where $T$ is the set of IoU thresholds and $C$ is the number of classes. A prediction is considered correct if the predicted class matches the ground truth and the IoU exceeds threshold $t$.

**Mean Recall for gDSA ($\text{mR}_g$).** In the gDSA task, we employ Mean Recall (mR) to evaluate the model's ability to detect relations, especially given multiple coexisting relations and class imbalance. To compute the mR, we first match predicted instances to ground truth based on class labels and Intersection over Union (IoU) with a threshold commonly set at 0.5. Next, we extract relations from the matched instances, defined as subject-object-prediction triplets. We then apply a relation confidence threshold $T_R$ and consider only relations with confidence scores above $T_R$. For each relation category $r$, the recall is computed as:

$$\text{Recall}r = \frac{\text{TP}r}{\text{TP}r + \text{FN}r}, \tag{7}$$

where TP$r$ is the number of true positives and FN$r$ is the number of false negatives for relation $r$. The Mean Recall is then calculated by averaging the recalls over all relation categories:

$$\text{mR} = \frac{1}{R} \sum_{r=1}^{R} \text{Recall}_r, \tag{8}$$

where $R$ is the total number of relation categories.

**Mean Average Precision for gDSA ($\text{mAP}_g$).** To further comprehensively assess model performance in document relational graph prediction, we use the Mean Average Precision for gDSA ($\text{mAP}_g$). We begin by performing instance matching and relation extraction as described in the computation of mR. We then evaluate the relations at confidence thresholds $T_R \in \{0.5, 0.75, 0.95\}$. For each relation category, we compute precision and recall, and calculate the Average Precision (AP) by integrating the precision-recall curve. The $\text{mAP}_g$ is then obtained by averaging the APs over all relation categories:

$$\text{mAP}_g = \frac{1}{R} \sum_{r=1}^{R} \text{AP}_r, \tag{9}$$

where $\text{AP}_r$ is the Average Precision for relation category $r$, and $R$ is the total number of relation categories. This metric balances precision and recall, rewarding models that predict correct relations with high confidence.

Since elements can have multiple relations, we treat relation prediction as a multi-label classification problem for each pair of instances. By evaluating performance per relation category and averaging, we ensure that rare but important relations are appropriately weighted, effectively addressing class imbalance. Additionally, relation evaluation depends on correctly detected instances, linking the quality of relation prediction to the performance on the DLA task. By employing mAP for the DLA task and $mR_g$ and $mAP_g$ for the gDSA task, we provide a comprehensive evaluation framework that addresses the challenges of document structure analysis, including multiple relations and class imbalance. This approach encourages the development of models capable of effectively interpreting complex document structures.

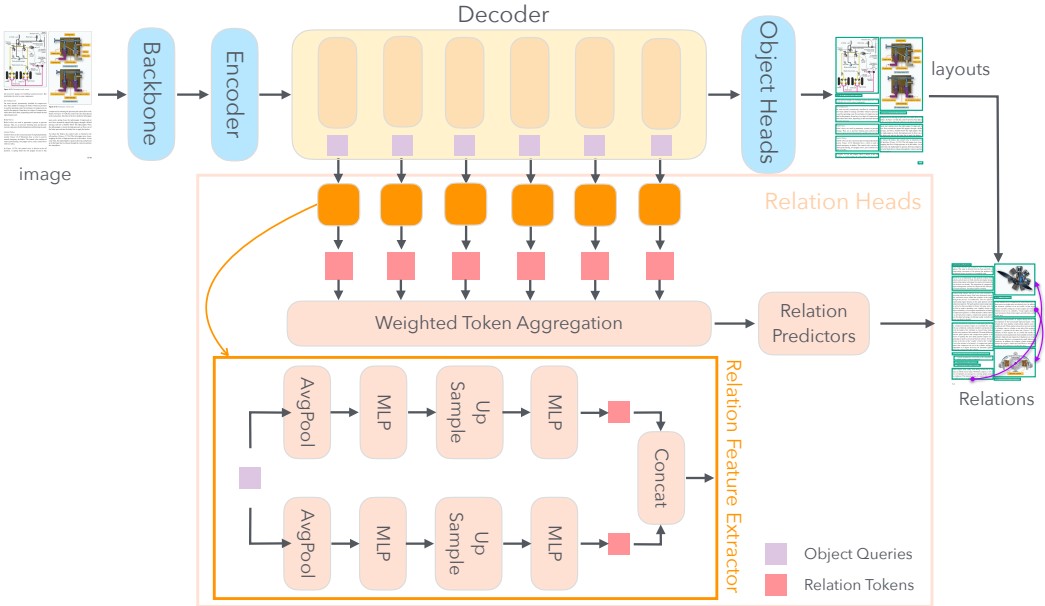

Figure 7: The overall architecture and the work flow of the proposed DRGG model. Given an image of document as input, the backbone will extract the feature from the document image and forward to the Encoder-Decoder architecture. The output of Decoder will be forwarded to the object heads and the relation heads for the prediction of document layouts and relations.

## C   DRGG

In this subsection, we provide a detailed structural analysis of the Document Relation Graph Generator (DRGG) and a detailed structural illustration as shown in Figure 7.

### C.1   ANALYSIS OF WEIGHTED TOKEN AGGREGATION STRATEGY

The Weighted Token Aggregation strategy in DRGG is a crucial mechanism that fine-tunes the importance of relational features extracted from different decoder layers, resulting in more accurate and refined predictions. In the DETR framework, object queries at various layers capture feature information at different scales and abstraction levels, which leads to inherent variations in the corresponding relational features. These differences are key to understanding how document elements relate to each other. Different types of relations in documents require attention to distinct aspects of the layout. For instance, *reference* relations requires a deeper focus on the content within the document elements. On the other hand, spatial relations demand more emphasis on the geometric properties and boundaries of the document elements. This nuanced understanding of relational features is what enables DRGG to employ a single relation head to effectively capture and classify multiple types of relations simultaneously. By adjusting the contribution of relational information from different decoder layers, DRGG can adapt to the varying scopes and demands of each type of relation, ensuring a comprehensive and precise representation of document structure.

### C.2   RELATION PREDICTOR WITH AUXILIARY RELATION HEAD

To enhance the stability and accuracy of DRGG's relational predictions, we introduce an auxiliary relation prediction head. This auxiliary relation head focuses solely on determining whether a relation exists between two document elements, without classifying the type of relation. By decoupling the existence of a relation from its categorization, the auxiliary relation head acts as a stabilizer, ensuring that false positives are minimized during inference.

During training, both the main relation predictor and the auxiliary relation head are trained simultaneously using Binary Cross Entropy (BCE) loss. At test time, the predictions from the auxiliary relation head are combined with the main relation predictor's output by multiplying their respec-

tive results. This multiplicative correction reduces uncertainty and enhances the robustness of the relational predictions.

Let the output from the main relation predictor, responsible for classifying specific relations, be denoted as $G_{\text{pred}} \in \mathbb{R}^{N \times N \times k}$, where $N$ represents the number of document elements and $k$ is the number of relation categories. Similarly, let the auxiliary relation head output, which predicts the existence of any relation between elements, be denoted as $A_{\text{pred}} \in \mathbb{R}^{N \times N}$, where each entry in $A_{\text{pred}}$ represents a binary prediction (relation exists or not) for a pair of document elements.

During inference, the final relational prediction $G_{\text{final}}$ is computed by multiplying the two outputs element-wise:

$$G_{\text{final}} = G_{\text{pred}} \odot A_{\text{pred}}^{\otimes k}, \tag{10}$$

where $\odot$ denotes the element-wise product, and $A_{\text{pred}}^{\otimes k}$ represents the auxiliary relation head's predictions expanded along the third dimension to match the number of relation categories $k$. This operation ensures that only relations that are confidently predicted to exist by the auxiliary relation head are retained in the final output.

### C.3 LOSS FUNCTION WITH HUNGARIAN MATCHING

For training, the loss computation in DRGG leverages the results of the **Hungarian matching algorithm** (Kuhn, 2010) from the object detection head in the final decoder layer. This algorithm ensures instance-level matching between predicted document elements and the ground truth elements, providing a one-to-one mapping between predictions and annotations. Once this matching is established, the predicted relation graph can be filtered and adjusted according to the matched pairs, which is critical for accurately training the relation predictor.

The Hungarian matching algorithm aims to minimize the total matching cost by finding the optimal permutation $\sigma^*$ that maps the set of predicted elements $\mathcal{P} = \{p_1, p_2, \ldots, p_N\}$ to the ground truth elements $\mathcal{T} = \{t_1, t_2, \ldots, t_N\}$. The cost function is defined as:

$$\text{Cost}(\sigma) = \sum_{i=1}^{N} \mathcal{L}(p_i, t_{\sigma(i)}), \tag{11}$$

where $\mathcal{L}(p_i, t_{\sigma(i)})$ is the loss between the predicted element $p_i$ and its matched ground truth element $t_{\sigma(i)}$. The optimal matching is obtained by minimizing this cost:

$$\sigma^* = \arg \min_{\sigma \in \mathfrak{S}_N} \sum_{i=1}^{N} \mathcal{L}(p_i, t_{\sigma(i)}), \tag{12}$$

where $\mathfrak{S}_N$ is the set of all possible permutations of $N$ elements. This matching is critical for aligning predicted relations with the ground truth during training, ensuring that predictions are corrected for each element's actual match.

The loss function for both the relation predictor and the auxiliary relation head is based on Binary Cross Entropy (BCE), computed independently for each of the predictions. Specifically, let $G_{\text{gt}} \in \mathbb{R}^{N \times N \times k}$ denote the ground truth relational graph, and let $A_{\text{gt}} \in \mathbb{R}^{N \times N}$ denote the ground truth existence of relations (i.e., whether a relation exists between pairs of elements). The total loss $\mathcal{L}_{\text{total}}$ is the sum of the losses for objects heads and relation predictor and the auxiliary relation head:

$$\mathcal{L}_{\text{total}} = \mathcal{L}_{\text{cls}} + \mathcal{L}_{\text{bbox}} + \lambda \mathcal{L}_{\text{rel}} + \sigma \mathcal{L}_{\text{rel}_{\text{aux}}}, \tag{13}$$

where $\lambda$ is a hyperparameter that controls the weight of the prediction head loss and $\sigma$ is another hyperparameter that controls the weight of the auxiliary relation head loss.

The relation prediction loss $\mathcal{L}_{\text{rel}}$ is defined as:

$$\mathcal{L}_{\text{rel}} = -\sum_{i,j=1}^{N} \sum_{c=1}^{K} \left( G_{\text{gt}}^{(i,j,k)} \log G_{\text{pred}}^{(i,j,k)} + (1 - G_{\text{gt}}^{(i,j,k)}) \log(1 - G_{\text{pred}}^{(i,j,k)}) \right), \qquad (14)$$

where $G_{\text{gt}}^{(i,j,k)}$ and $G_{\text{rel}}^{(i,j,k)}$ denote the ground truth and predicted probabilities for the $k$-th relation category between elements $i$ and $j$.

Similarly, the auxiliary relation existence loss $\mathcal{L}_{\text{aux}}$ is given by:

$$\mathcal{L}_{\text{rel}_{\text{aux}}} = -\sum_{i,j=1}^{N} \left( A_{\text{gt}}^{(i,j)} \log A_{\text{pred}}^{(i,j)} + (1 - A_{\text{gt}}^{(i,j)}) \log(1 - A_{\text{pred}}^{(i,j)}) \right). \qquad (15)$$

By incorporating both losses, DRGG is trained to accurately predict both the existence and the type of relations between document elements. The auxiliary relation head plays a crucial role in stabilizing the predictions, while the Hungarian matching ensures precise, instance-level alignment between predictions and ground truth, thus improving the overall quality of the relational graph.

## D   ADDITIONAL RESULTS OF DRGG

In this section, we provide detailed supplementary results from our additional DRGG experiments to offer deeper insights into the gDSA task and the structural design of DRGG.

**Results on Different Document Domains of GraphDoc Dataset.**

To comprehensively evaluate the performance of DRGG, we conducted experiments across multiple document domains separately in GraphDoc dataset, reflecting diverse layouts and structural complexities. These experiments aim to demonstrate the adaptability of our method to varying document types. The detailed results of six different document domains (i.e., Financial Reports, Scientific Articles, Laws and Regulations, Government Tenders, Manuals, and Patents) are presented in Table 4 and Table 5 below. We used InternImage as the backbone, RoDLA as the detector, and DRGG for relationship extraction. The tables below summarize the performance in terms of mRg and mAPg under relation confidence thresholds of 0.5, 0.75, and 0.95 under the IoU threshold of 0.5.

Table 4: $\text{mR}_g$ Results on different document domains of GraphDoc Dataset.

| Relation Confidence Thresholds | Financial Reports | Scientific Articles | Laws and Regulations | Government Tenders | Manuals | Patents |
|---|---|---|---|---|---|---|
| 0.5 | 15.0 | 46.3 | 38.7 | 40.6 | 40.6 | 22.7 |
| 0.75 | 12.3 | 42.0 | 36.5 | 38.7 | 35.6 | 20.5 |
| 0.95 | 9.0 | 35.6 | 33.5 | 34.1 | 27.1 | 17.5 |

Table 5: $\text{mAP}_g$ Results on different document domains of GraphDoc dataset.

| Relation Confidence Thresholds | Financial Reports | Scientific Articles | Laws and Regulations | Government Tenders | Manuals | Patents |
|---|---|---|---|---|---|---|
| 0.5 | 52.6 | 54.5 | 63.2 | 55.9 | 46.8 | 31.8 |
| 0.75 | 50.9 | 52.9 | 58.7 | 51.4 | 44.4 | 30.7 |
| 0.95 | 20.2 | 47.5 | 54.6 | 48.1 | 32.5 | 29.3 |

The results demonstrate clear domain-specific trends. Laws and Regulations achieve the highest $\text{mAP}_g$@0.5 with 63.2, benefiting from their structured and consistent layouts, while Patents perform worst, with $\text{mR}_g$@0.95 at 17.5, due to their dense and complex layouts. Both $\text{mR}_g$ and $\text{mAP}_g$ decline as the relation confidence threshold increases, reflecting the challenges of capturing precise relationships under stricter criteria. These findings highlight the varying complexities across domains and the need for robustness in handling diverse document structures.

**Results on Spatial and Logical Relations of GraphDoc Dataset.**

To investigate the impact of different relationship types, we analyzed DRGG's performance on documents containing only spatial relations compared to those containing both spatial and logical relations. We used InternImage as the backbone, RoDLA as the detector, and DRGG for relationship

prediction. We used InternImage as the backbone, RoDLA as the detector, and DRGG for relationship extraction. $mR_g$ and $mAP_g$ metrics were computed under relation confidence thresholds of 0.5, 0.75, and 0.95 with an IoU threshold of 0.5, as shown in Table 6.

Table 6: Results for relation prediction performance under different relation types.

| Spatial Relation | Logical Relation | $mR_g$@0.5 | $mR_g$@0.75 | $mR_g$@0.95 | $mAP_g$@0.5 | $mAP_g$@0.75 | $mAP_g$@0.95 |
|---|---|---|---|---|---|---|---|
| √ | | 32.1 | 27.7 | 22.1 | 49.5 | 49.5 | 41.3 |
| √ | √ | 26.7 | 23.9 | 20.1 | 57.5 | 56.2 | 37.6 |

The results show that capturing spatial and logical relations is challenging, as indicated by the lower metrics. Spatial relations alone achieve an $mR_g$@0.5 of 32.1 and a $mAP_g$@0.5 of 49.5. When logical relations are included, $mR_g$@0.5 drops to 26.7, while $mAP_g$@0.5 slightly improves to 57.5. Nevertheless, performance declines significantly at stricter thresholds, i.e., $mR_g$@0.95 and $mA_g$@0.95.

# E    ABLATION STUDY RESULT OF DRGG

In this section, we present the ablation study of the DRGG design to validate the effectiveness of the DRGG model. The analysis evaluates four key aspects: the impact of using DRGG as a relational graph prediction head, the effectiveness of the relation feature extractor module, the influence of IoU thresholds, and the effect of relation confidence thresholds on different relation types.

**Ablation of DRGG Model.** Table 7 highlight the effectiveness of integrating the DRGG relation prediction head into the document layout analysis task. InternImage combined with DINO sees an improvement from 80.5 to 81.5, the highest among all configurations, illustrating the harmony between the DRGG head and advanced backbones. This improvement mark the DRGG module's utility in capturing complex document structures, as it effectively augments the detector's ability to model relationships between document elements. These findings validate the design of DRGG and its critical role in advancing the accuracy and reliability of document structure analysis.

Table 7: Ablation study of DRGG model impact for DLA Task

| Backbone | Detector | Relation Head | DLA mAP@50:5:95 |
|---|---|---|---|
| InternImage | DINO | | 76.6 |
| ResNet | | | 74.3 |
| ResNeXt | RoDLA | - | 77.7 |
| InternImage | | | 80.5 |
| InternImage | DINO | | 79.5 |
| ResNet | | | 71.0 |
| ResNeXt | RoDLA | DRGG | 77.9 |
| InternImage | | | **81.5** |

**Ablation of Relation Feature Extractor.** Table 8 illustrates the importance of the relation feature extractor in the DRGG model. When paired with InternImage and RoDLA, the feature extractor significantly outperforms a linear layer replacement across all metrics. For DLA, it achieves a higher mAP result of 81.5. In gDSA, the extractor shows clear advantages in $mR_g$ and $mAP_g$.

Table 8: Ablation study of relation feature extractor module in DRGG model compared with single linear layer instead of relation feature extractor module in DRGG model

| Backbone | Detector | Relation Head | DLA mAP@50:5:95 | gDSA $mR_g$@0.5 | $mAP_g$@0.5 | $mAP_g$@0.75 | $mAP_g$@0.95 |
|---|---|---|---|---|---|---|---|
| InternImage | RoDLA | DRGG | 81.5 | 30.7 | 57.6 | 56.3 | 46.5 |
| | | linear layer in DRGG | 79.9 | 25.8 | 52.9 | 42.3 | 30.5 |

**Ablation of IoU Thresholds.** We understand the importance of evaluating model performance under high IoU thresholds to assess alignment between predicted and actual bounding boxes. To evaluate the impact of high IoU thresholds on model performance, we conducted experiments using InternImage as the backbone, RoDLA as the detector, and DRGG for relationship extraction. The results of Table 9 below present $mR_g$ and $mAP_g$ values under IoU thresholds of 0.5, 0.75, and 0.95:

Table 9: Impact of IoU thresholds on $mR_g$ and $mAP_g$.

| IoU Threshold | $mR_g$@0.5 | $mR_g$@0.75 | $mR_g$@0.95 | $mAP_g$@0.5 | $mAP_g$@0.75 | $mAP_g$@0.95 |
|---|---|---|---|---|---|---|
| 0.5 | 30.7 | 28.2 | 24.5 | 57.6 | 56.3 | 46.5 |
| 0.75 | 28.8 | 26.5 | 23.0 | 56.7 | 54.8 | 36.8 |
| 0.95 | 22.1 | 20.7 | 18.4 | 55.5 | 54.3 | 36.5 |

As shown in the results, at the highest IoU threshold of 0.95, the model achieves 18.4 $mR_g$@0.95 and 36.5 $mAP_g$@0.95, demonstrating the significant challenges in capturing precise alignments, particularly in complex or densely packed layouts where bounding box prediction errors have a greater impact. While lower IoU thresholds allow the model to achieve higher recall and precision, stricter thresholds demand fine-grained alignment, which may not always be feasible due to the inherent limitations of bounding box prediction accuracy. These findings emphasize the need to balance strict alignment metrics with practical utility based on specific application requirements. Higher IoU thresholds, while providing stricter metrics, may not fully capture the model's overall effectiveness in scenarios where moderate overlap suffices.

**Ablation of Relation Confidence Thresholds among Relation Categories.** Table 10 and Table 11 shows the influence of different relationship confidence thresholds in the context of imbalanced sample sizes among relation categories. We used InternImage as the backbone, RoDLA as the detector, and DRGG for relationship extraction. mRg @0.5, mRg @0.75, and mRg @0.95 denote the mean Recall in the gDSA Task for relation confidence threshold 0.5, 0.75, and 0.95 under IoU threshold 0.5, respectively. mAPg @0.5, mAPg @0.75, and mAPg @0.95 denote the mean Average Precision in the gDSA Task for relation confidence threshold 0.5, 0.75, and 0.95 under IoU threshold 0.5, respectively.:

Table 10: $mR_g$ Results at different relation confidence thresholds.

| Confidence Threshold | Up | Down | Left | Right | Parent | Child | Sequence | Reference |
|---|---|---|---|---|---|---|---|---|
| 0.5 | 41.7 | 50.0 | 71.4 | 71.4 | 12.5 | 25.0 | 0.0 | 0.0 |
| 0.75 | 41.7 | 33.3 | 42.9 | 57.1 | 12.5 | 12.5 | 0.0 | 0.0 |
| 0.95 | 8.3 | 8.3 | 28.6 | 28.6 | 12.5 | 0.0 | 0.0 | 0.0 |

Table 11: $mAP_g$ Results at different relation confidence thresholds.

| Confidence Threshold | Up | Down | Left | Right | Parent | Child | Sequence | Reference |
|---|---|---|---|---|---|---|---|---|
| 0.5 | 49.0 | 49.0 | 99.0 | 99.0 | 45.5 | 45.5 | 56.4 | 16.8 |
| 0.75 | 47.4 | 45.1 | 99.0 | 99.0 | 45.5 | 45.5 | 51.2 | 16.8 |
| 0.95 | 40.4 | 40.4 | 49.5 | 49.5 | 37.6 | 36.6 | 46.5 | 0.0 |

From the experiment result, we could find that, spatial relations, i.e., *Left*, *Right*, *Up*, and *Down* achieve consistently higher mR and mAP values compared to logical relations, i.e., *Parent*, *Child*, *Sequence*, and *Reference*, reflecting their prevalence in the dataset and larger training sample sizes. As the confidence threshold increases, both mR and mAP values decline across all relation types, with logical relations showing the steepest drop; for instance, Reference achieves 16.8 mAP at a 0.5 threshold but drops to 0.0 at 0.95, highlighting the challenges of capturing infrequent or ambiguous relationships. A confidence threshold of 0.5 strikes a balance between precision and recall, but addressing dataset imbalance through weighted training could further enhance performance.

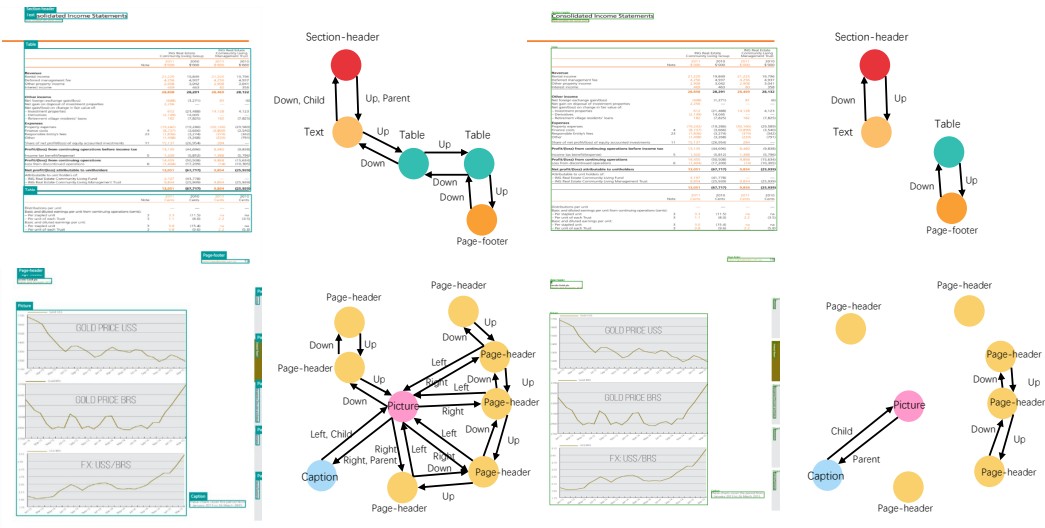

Figure 8: Qualitative Results for DRGG prediction, compared with ground truth on GraphDoc Dataset.

# F IMPLEMENTATION DETAILS

**Hardware Setup.** In this work, all experiments were conducted on a computing cluster node equipped with four Nvidia A100 GPUs, each with 40 GB of memory. Each node would also with 300 GB of CPU memory.

**Training Settings.** We implemented our method using PyTorch v1.10 and trained the model with the AdamW optimizer using a batch size of 4. The initial learning rate was set to $1 \times 10^{-4}$, with a weight decay of $5 \times 10^{-3}$. The AdamW hyperparameters, betas and epsilon, were configured to $(0.9, 0.999)$ and $1 \times 10^{-8}$, respectively. To enhance the model's robustness and accuracy, we employed a multi-scale training strategy. Specifically, the shorter side of each input image was randomly resized to one of the following lengths: 480, 512, 544, 576, 608, 640, 672, 704, 736, 768, 800, while ensuring that the longer side did not exceed 1333 pixels. This approach helps the model generalize better to varying document sizes and layouts, reflecting the diverse nature of real-world document data.

# G QUALITATIVE RESULTS OF DRGG

In this section, we present several qualitative results predicted by DRGG on GraphDoc validation dataset, alongside their corresponding ground truth annotations for comparison.

As illustrated in Figure 8, errors in relation prediction arise primarily from two sources. First is the ambiguity in densely populated layouts, where elements, e.g., captions and figures, lack clear alignment. Secondly, misclassification and inaccurate bounding boxes, from the DLA stage, propagate errors to the relation prediction process. Despite these challenges, DRGG demonstrates promising capabilities in capturing key spatial and logical relationships, such as parent-child links between *Picture* and *Caption*. Nonetheless, the DRGG performance is hindered in DLA accuracy, as seen in cases of misclassified tables leading to missing relationships. To address these issues, we suggest incorporating multimodal embeddings that combine visual and textual features, improving the DLA backbone for enhanced detection performance, and integrating post-processing methods to refine predictions using contextual cues. Additionally, extending DRGG to multi-page relational understanding will enhance its applicability for comprehensive document structure analysis and relation predictions.

