# OpenReview forum: "Graph-based Document Structure Analysis"
_ICLR.cc/2025/Conference — ICLR 2025 Poster_

### Official Review · Reviewer_G241 · 2024-10-27

**Soundness:** 3
**Presentation:** 4
**Contribution:** 3
**Rating:** 8
**Confidence:** 4

**Summary:**

In this paper, the authors introduced an approach to analyzing document structures using graph theory, which offers a unique perspective on understanding the complex relationships and patterns within documents. The primary contribution of this paper lies in its methodology that transforms textual content into graphical representations, enabling researchers and practitioners to visualize and analyze the hierarchical and relational aspects of documents more effectively.

**Strengths:**

1. The authors propose a method that represents documents as graphs, where nodes could represent sentences or paragraphs and edges could capture relationships between these elements based on semantic similarity or other linguistic features. This original perspective offers a fresh way of understanding the hierarchical organization within texts.

2. The authors provide a clear and systematic approach to converting textual documents into graph structures, followed by an analysis that leverages graph algorithms. They also discuss the advantages of their method over traditional text-based approaches, such as improved handling of complex document structures and enhanced interpretability through visual representations.

3.  The clarity of the paper is good. The authors have structured the content logically, starting with an introduction to graph theory and its relevance to document analysis, followed by a detailed explanation of their method, implementation details, and results.

**Weaknesses:**

1. The paper could benefit from a more explicit discussion on potential weaknesses or limitations of the proposed method. This includes considerations such as scalability with large datasets, robustness to noise, and applicability across different types of documents.

2. There are lack of comparison with MLLM based methods. The results of GPT4 should be reported for readers to compare.

**Questions:**

1. A higher IoU score suggests a better alignment between the predicted and actual bounding boxes.What is the IoU score when it falls between 0.7 - 0.95  and  0.8 - 0.95?

---

> ### Author Response · Authors · 2024-11-26
> **Response to Reviewer G241**
>
> **\[W1\] Discussion on limitations**
>
> > The paper could benefit from a more explicit discussion on potential weaknesses or limitations of the proposed method. This includes considerations such as scalability with large datasets, robustness to noise, and applicability across different types of documents.
>
> Thank you for highlighting the importance of discussing the potential weaknesses and limitations of our method. We acknowledge that scalability with large datasets is crucial, and we plan to validate our approach on larger datasets in future work. Regarding robustness to noise, we intend to follow the RoDLA paper setting to test our method's robustness. As for applicability across different types of documents, our dataset already includes five diverse domains, but we recognize the need for broader applicability and are working towards that. We have discussed these limitations in Section 5 of our conclusion and will expand on them in the supplementary to provide a clearer understanding of the scope and potential areas for improvement in our work.
>
> ---
>
> **\[W2\] Comparison with MLLM based methods**
>
> > There are lack of comparison with MLLM based methods. The results of GPT4 should be reported for readers to compare.
>
> Thank you for highlighting the need for comparisons with general Multimodal Large Language Model (MLLM) based methods. We have incorporated experimental results to provide a comprehensive comparison, including evaluations of general models such as **LLaVA-OneVision-7B**, **Qwen2-VL-Instruct-7B**, and **Pixtral-12B** on Document Layout Analysis (DLA) tasks. Below are the results:
>
> **DLA results**
>
> | MLLM Model           | **mAP@50:5:95** |
> | -------------------- | --------------- |
> | LLaVA-OneVision-7B   | $1.03e^{-5}$        |
> | Qwen2-VL-Instruct-7B | $3.33e^{-5}$        |
> | Pixtral-12B          | $0.0$             |
>
> We agree that incorporating MLLM-based methods into the comparison is a valuable idea, but unfortunately, the results were not promising. As shown in the tables, all tested MLLM methods exhibit nearly zero performance across the evaluated metrics in DLA tasks. We have tried different prompts with MLLM, and also different post-processing methods to extract the detection result from the generated outputs from the MLLMs. Nonetheless, we found only very few document elements could be correctly recognized.
>
> These results reflect the current limitations of general MLLMs in handling document layout analysis tasks, which demand explicit structural understanding rather than general multimodal reasoning capabilities.
>
> ---
>
> **\[Q1\]Model Performance under High IoU Requirements**
>
> > A higher IoU score suggests a better alignment between the predicted and actual bounding boxes. What is the IoU score when it falls between 0.7 - 0.95 and 0.8 - 0.95?
>
> We understand the importance of evaluating model performance under high IoU thresholds to assess alignment between predicted and actual bounding boxes. To evaluate the impact of high IoU thresholds on model performance, we conducted experiments using InternImage as the backbone, RoDLA as the detector, and DRGG for relationship extraction. The results below present **mRg** and **mAPg** values under IoU thresholds of 0.5, 0.75, and 0.95:
>
> | IoU threshold | **mRg@0.5** | **mRg@0.75** | **mRg@0.95** | **mAPg@0.5** | **mAPg@0.75** | **mAPg@0.95** |
> | ------------- | ----------- | ------------ | ------------ | ------------ | ------------- | ------------- |
> | 0.5           | 30.7        | 28.2         | 24.5         | 57.6         | 56.3          | 46.5          |
> | 0.75          | 28.8        | 26.5         | 23.0         | 56.7         | 54.8          | 36.8          |
> | 0.95          | 22.1        | 20.7         | 18.4         | 55.5         | 54.3          | 36.5          |
>
> As shown in the table results, at the highest IoU threshold of 0.95, the model achieves 18.4 mRg@0.95 and 36.5 mAPg@0.95, demonstrating the significant challenges in capturing precise alignments, particularly in complex or densely packed layouts where bounding box prediction errors have a greater impact. While lower IoU thresholds allow the model to achieve higher recall and precision, stricter thresholds demand fine-grained alignment, which may not always be feasible due to the inherent limitations of bounding box prediction accuracy. These findings emphasize the need to balance strict alignment metrics with practical utility based on specific application requirements. Higher IoU thresholds, while providing stricter metrics, may not fully capture the model's overall effectiveness in scenarios where moderate overlap suffices. This analysis has been included in the revised paper to address the reviewer's concerns.

---

### Official Review · Reviewer_zRCK · 2024-10-29

**Soundness:** 3
**Presentation:** 3
**Contribution:** 2
**Rating:** 6
**Confidence:** 4

**Summary:**

The paper defines a graph-based Document Structure Analysis task (gDSA) and builds a new Document Structure Analysis dataset, GraphDoc, based on existing datasets. By constructing a document layout graph structure that includes spatial and logical relationships, it enhances the depth of document structure understanding. Additionally, the paper presents a new end-to-end Document Relation Extraction model, DRGG, designed to extract features from the document layout and generate a relation graph that includes spatial and logical relationships among document elements.

**Strengths:**

The author compares the proposed DRGG framework with several existing document layout analysis and graph structure analysis methods, demonstrating through experiments that this model exhibits outstanding performance on the gDSA task, particularly in handling complex document structures, which is significant for achieving deeper document understanding.

**Weaknesses:**

1. Most document instances exhibit multiple relationships. The paper should provide additional experiments to evaluate the precision and recall of DRGG in capturing instances that have both spatial and logical relationships, compared to those with only one type of relationship.
2. In the dataset, the number of samples in the position relationship category is significantly higher than that in the logical relationship category. It would be better to consider adding experiments to demonstrate how to select the relationship confidence threshold to ensure evaluation accuracy in the context of imbalanced sample sizes among relationship categories.

**Questions:**

1. A performance bottleneck has been observed in detecting reference relationships in the experiments. Given the importance of reference relationships in document structure understanding, could you supplement with an analysis of the error cases in reference relationship detection, discussing the reasons for these errors and suggesting improvements.
2. It is suggested that the author can elaborate on the utility of the proposed dataset, evaluation tasks, and evaluation metrics in understanding complex document interactions. Is it possible to provide specific examples of how the GraphDoc dataset and DRGG model can be applied in real-world document understanding tasks? Highlighting of these advantages, such as Reading Order Prediction and Document Hierarchical Structure Analysis tasks mentioned in the paper.

---

> ### Author Response · Authors · 2024-11-26
> **Response to Reviewer zRCK - Part 1**
>
> We thank Reviewer zRCK for your valuable feedback and comments. We appreciate your comments on the human modification of relationships. We will address your concerns and questions in the response below.
>
> **\[W1\] Comparision between different elements**
>
> > Most document instances exhibit multiple relationships. The paper should provide additional experiments to evaluate the precision and recall of DRGG in capturing instances that have both spatial and logical relationships, compared to those with only one type of relationship.
>
> In our dataset, spatial relationships are more prevalent, and no documents were found with only logical relationships and no spatial ones. Thus, we evaluated DRGG’s performance on documents with only spatial relationships versus those with both spatial and logical relationships. We used InternImage as the backbone, RoDLA as the detector, and DRGG for relationship extraction. mRg @0.5, mRg @0.75, and mRg @0.95 denote the mean Recall in the gDSA Task for relation confidence threshold 0.5, 0.75, and 0.95 under IoU threshold 0.5, respectively. mAPg @0.5, mAPg @0.75, and mAPg @0.95 denote the mean Average Precision in the gDSA Task for relation confidence threshold 0.5, 0.75, and 0.95 under IoU threshold 0.5, respectively.
>
> | **Spatial Relation** | **Logical Relation** | **mRg@0.5** | **mRg@0.75** | **mRg@0.95** | **mAPg@0.5** | **mAPg@0.75** | **mAPg@0.95** |
> | -------------------- | -------------------- | ----------- | ------------ | ------------ | ------------ | ------------- | ------------- |
> | ✔                    |                      | 32.1        | 27.7         | 22.1         | 49.5         | 49.5          | 41.3          |
> | ✔                    | ✔                    | 26.7        | 23.9         | 20.1         | 57.5         | 56.2          | 37.6          |
>
> The results indicate that capturing spatial and logical relationships is more challenging, reflected in the lower performance. This analysis have been included in the revised paper to address the reviewer’s concern.

---

> ### Author Response · Authors · 2024-11-26
> **Response to Reviewer zRCK - Part 2**
>
> **\[W2\] Influence of different relation confidence threshold**
>
> > In the dataset, the number of samples in the position relationship category is significantly higher than that in the logical relationship category. It would be better to consider adding experiments to demonstrate how to select the relationship confidence threshold to ensure evaluation accuracy in the context of imbalanced sample sizes among relationship categories.
>
> Thank you very much for your suggestions. We have added analysis with  the influence of different relationship confidence thresholds in the context of imbalanced sample sizes among relationship categories. We used InternImage as the backbone, RoDLA as the detector, and DRGG for relationship extraction. mRg @0.5, mRg @0.75, and mRg @0.95 denote the mean Recall in the gDSA Task for relation confidence threshold 0.5, 0.75, and 0.95 under IoU threshold 0.5, respectively. mAPg @0.5, mAPg @0.75, and mAPg @0.95 denote the mean Average Precision in the gDSA Task for relation confidence threshold 0.5, 0.75, and 0.95 under IoU threshold 0.5, respectively.
>
> **mRg Results (Mean Recall for relational graph)**
>
> | **Confidence Threshold** | **Up** | **Down** | **Left** | **Right** | **Parent** | **Child** | **Sequence** | **Reference** |
> | ------------------------ | ------ | -------- | -------- | --------- | ---------- | --------- | ------------ | ------------- |
> | **0.5**                  | 41.7   | 50.0     | 71.4     | 71.4      | 12.5       | 25.0      | 0.0          | 0.0           |
> | **0.75**                 | 41.7   | 33.3     | 42.9     | 57.1      | 12.5       | 12.5      | 0.0          | 0.0           |
> | **0.95**                 | 8.3    | 8.3      | 28.6     | 28.6      | 12.5       | 0.0       | 0.0          | 0.0           |
>
> **mAPg Results (Mean Average Precision for relational graph)**
>
> | **Confidence Threshold** | **Up** | **Down** | **Left** | **Right** | **Parent** | **Child** | **Sequence** | **Reference** |
> | ------------------------ | ------ | -------- | -------- | --------- | ---------- | --------- | ------------ | ------------- |
> | **0.5**                  | 49.0   | 49.0     | 99.0     | 99.0      | 45.5       | 45.5      | 56.4         | 16.8          |
> | **0.75**                 | 47.4   | 45.1     | 99.0     | 99.0      | 45.5       | 45.5      | 51.2         | 16.8          |
> | **0.95**                 | 40.4   | 40.4     | 49.5     | 49.5      | 37.6       | 36.6      | 46.5         | 0.0           |
>
> From the experiment result, we could find that, spatial relations, i.e.,  _Left_, _Right_, _Up_, and _Down_ achieve consistently higher mR and mAP values compared to logical relations, i.e., _Parent_, _Child_, _Sequence_, and _Reference_, reflecting their prevalence in the dataset and larger training sample sizes. As the confidence threshold increases, both mR and mAP values decline across all relation types, with logical relations showing the steepest drop; for instance, _Reference_ achieves 16.8 mAP at a 0.5 threshold but drops to 0.0 at 0.95, highlighting the challenges of capturing infrequent or ambiguous relationships. A confidence threshold of 0.5 strikes a balance between precision and recall, but addressing dataset imbalance through weighted training could further enhance performance.
>
> **\[Q1\] Error discussion and improvements**
>
> > A performance bottleneck has been observed in detecting reference relationships in the experiments. Given the importance of reference relationships in document structure understanding, could you supplement with an analysis of the error cases in reference relationship detection, discussing the reasons for these errors and suggesting improvements.
>
> We acknowledge that relation graph extraction poses challenges in document structure analysis, especially based onthe  pure vision aspect. From our analysis in Supplementary Section  **F Qualitative Results of DRGG**, errors in relation prediction are primarily due to two factors:
>
> 1. Ambiguity in dense layouts where elements like captions and figures are not clearly aligned, leading to missed or incorrect links
> 2. Errors from the Document Layout Analysis (DLA) stage, such as misclassification or wrong bounding boxes, propagate to DRGG and hinder accurate relation prediction.
>
> To address these issues, we suggest the following improvements:
>
> 1. Incorporating richer multimodal embeddings that integrate visual and textual features to enhance semantic understanding
> 2. Improving the DLA backbone for more accurate segmentation and classification of critical elements like captions and figures
> 3. Integrating post-processing methods, such as rule-based refinement, to verify and correct relations using contextual cues.
>
> These  enhancements will guide our future work and are critical for advancing relation prediction. We have included this analysis and these suggestions in the revised version to better improve our work.

---

> ### Author Response · Authors · 2024-11-26
> **Response to Reviewer zRCK - Part 3**
>
> **\[Q2\] Specific examples of applications on real-world document understanding tasks**
>
> > It is suggested that the author can elaborate on the utility of the proposed dataset, evaluation tasks, and evaluation metrics in understanding complex document interactions. Is it possible to provide specific examples of how the GraphDoc dataset and DRGG model can be applied in real-world document understanding tasks? Highlighting of these advantages, such as Reading Order Prediction and Document Hierarchical Structure Analysis tasks mentioned in the paper.
>
> The GraphDoc dataset and DRGG model provide significant utility in understanding complex document interactions, particularly by enabling advanced applications in real-world document understanding tasks as follows:
>
> 1. Semantic Reading Order Prediction: Unlike traditional linear-based reading order, the graph structure in GraphDoc provides a semantic reading order that incorporates both spatial and logical relationships. For example, in reference-heavy documents, such as research papers or financial reports, the graph can establish meaningful connections between captions, figures, and text references, enabling more accurate interpretation and navigation of the document.
> 2. Document Structure Analysis: Graph-based representations are inherently more flexible than tree-based hierarchical structures. For instance, while tree structures could only represent strict parent-child relationships, graphs can capture more complex interactions, e.g., references across document sections. This makes the GraphDoc dataset and DRGG especially useful for real-world tasks, e.g., question answering (QA), where understanding the document's semantic structure is critical. For example, in a QA task, the graph structure enables the linking of a question about a figure to the corresponding caption, references, and related paragraphs, providing accurate and comprehensive answers.
> 3. Applications for Accessibility: For blind or visually impaired users, DRGG-powered systems can provide a guided experience through complex documents, where spatial and logical relationships, e.g., between tables, text, and images, are essential to understanding document content. By leveraging graph-based relation representation, such systems can deliver meaningful context rather than a fragmented linear description of the document.

---

### Official Review · Reviewer_HRLr · 2024-11-03

**Soundness:** 3
**Presentation:** 3
**Contribution:** 3
**Rating:** 8
**Confidence:** 4

**Summary:**

This paper introduces the **GraphDoc dataset**, a new dataset for document layout and structure analysis derived from the DocLayNet dataset by adding relation annotations. The authors propose a new task called **graph-based Document Structure Analysis (gDSA)** that involves not only detecting document elements, but also generating **spatial and logical relations** in the form of a graph. To address this new task, the authors also propose a model called the **Document Relation Graph Generator (DRGG)** that can generate relation graphs from document layouts, achieving 57.6% mAPg@0.5 on the GraphDoc dataset.  The authors hope that this graphical representation of document structure will be an innovative advancement in document structure analysis and understanding.

**Strengths:**

This is a large new dataset (80,000 single-page document images ; 1.10 million instances across 11 categories: Caption, Footnote, Formula, Listitem, Page-footer, Page-header, Picture, Section-header, Table, Text, and Title.) that extends DocLayNet Pfitzmann et al. (2022), and the relative location of the document elements is key to document understanding

**Weaknesses:**

This paper doesn't really have any major weaknesses. it is mostly a dataset paper and the dataset is strong. The authors also introduce a new model, which is rather straightforward but still interesting.

**Questions:**

Line 284: "most of the results have been manually verified and refined". What percentage of results have been manually verified and what percentage of results have been manually refined?

---

> ### Author Response · Authors · 2024-11-26
> **Response to Reviewer HRLr**
>
> We thank Reviewer HRLr for your valuable feedback and comments. We appreciate your comments on the human modification of relationships. We will address your concerns and questions in the response below.
>
> **\[Q1\] Percentage of human verification and refinement part**
>
> > Line 284: "most of the results have been manually verified and refined". What percentage of results have been manually verified and what percentage of results have been manually refined?
>
> We conducted extensive human verification and refinement, covering approximately 58.5% of the dataset. Specifically, we reviewed 4,852 pages of Government Tenders, 12,000 pages of Financial Reports, 6,469 pages of Patents, and 8,000 pages from other domains. The refinement rates for relation labels varied across domains, with 23% of relation labels refined in Financial Reports, 8% in Scientific Articles, 26% in Government Tenders, and 17% in Patents. These efforts ensured a high standard of quality in the annotated dataset.

---

### Official Review · Reviewer_jDzA · 2024-11-04

**Soundness:** 2
**Presentation:** 2
**Contribution:** 3
**Rating:** 6
**Confidence:** 4

**Summary:**

This paper introduces a dataset which is built on top of the DocLayNet dataset to provide detailed annotation of both spatial and logical relation annotations for understanding the structure of single-page documents. As well as, a new framework, Document Relation Graph Generator, is proposed as a baseline for conducting layout and structure understanding in an end-to-end manner.

**Strengths:**

1. The research problem is interesting and novel which is the less explored in this area.
2. The dataset generation and the evaluation metrics look reasonable.
3. Some analyses are conducted to show the insight of the datasets such as the correlations and co-occurrence of different relation pairs.

**Weaknesses:**

1. Some key concepts are not clearly defined: e.g. why spatial relations are important, why only four spatial relation types are defined, Is proper all documents from different domains to adopt the same rule to define the relations between semantic entities.
2. Lack of analysis about rule-based generated relation and human refining details which is essential for demonstrating the high quality of the dataset.
3. It would be clearer if the detailed categorisation of parent, child, and reference relation types could be provided with examples (visualised would be much better) to enable the reader to understand definitions and tasks clearly.
4. The description of Document Relation Graph Generator is not clearly described to me. The overall workflow with the corresponding aim is relatively hard to understand. I have gone through it several times but still hard to understand including input to the framework, why the encoder and encoder structure, and how the decoder works during training and inferencing.
5. The experimental section is underexplored. Analysis of different document domains, fine-grained relation type analysis (section->subsection, section->paragraph).
6. Similar to quantitative analysis, more qualitative analyses are expected.
I really like the idea of the paper, but more in-depth analysis is essential to maximising the contributions of the author desired.

**Questions:**

1. Why do you think spatial relation should be considered as one of the relation types of this dataset which might be directly acquired by leveraging the predicted bbox coordinated?
2. Could you provide more statistics details before and after human checking?
3. Do you think the rules might be updated based on different document domains?
4. For the relation prediction dataset, is there any reason you choose single-page datasets? It is more natural to use multipage-scenarios to establish the entire document's logical structure.

---

> ### Author Response · Authors · 2024-11-26
> **Response to Reviewer jDZA - Part 1**
>
> We thank Reviewer jDzA for the valuable feedback and comments on the relationship definition, analysis details, presentation, and models. We will address your concerns and questions below.
>
> ---
>
> **\[W1\] Clarification of key concepts**
>
> > Some key concepts are not clearly defined: e.g. why spatial relations are important, why only four spatial relation types are defined, Is proper all documents from different domains to adopt the same rule to define the relations between semantic entities.
>
> Spatial relations are essential in document structure analysis because they provide contextual information beyond the raw bounding boxes of document layout elements. Simply knowing the positions of elements is insufficient for understanding the document's relational structure, especially when real-world perturbations occur, e.g., document image rotation and translation. By defining four fundamental spatial relation types, we aim to capture how document elements interact within a document fundamentally, facilitating a more robust and generalized understanding across different domains. While documents from various domains may have unique characteristics, adopting a consistent and general rule of relations allows for a unified approach to structure analysis. To address domain-specific nuances and ensure accuracy, we incorporate human verification, which helps adapt our method to diverse document domains while maintaining relation type definition principles.
>
> ---
>
> **\[W2\] Lack of analysis of human refining details**
>
> > Lack of analysis about rule-based generated relation and human refining details which is essential for demonstrating the high quality of the dataset.
>
> We appreciate the reviewer's feedback regarding the analysis of rule-based generated relations and the details of human refinement, which are indeed essential for demonstrating the high quality of our dataset. In response to the previous W1, we adopted a cross-human verification process to correct any inaccuracies in the automatically generated relations. Six inspectors were involved in this task, with each spending an average of 100 working hours to refine the data content. For instance, in financial reports, the rules for logical relations differ from those in other document types. It is specifically needed to adjust the sequence relationships by isolating company seals or logos from the surrounding text to accurately reflect their distinct roles in the document structure. By incorporating such detailed refinements, we ensure the high quality and reliability of our new dataset. Besides, we plan to release the entire dataset in the future to support further research and development in this field.
>
> ---
>
> **\[W3\] Examples for categorization of relation types**
>
> > It would be clearer if the detailed categorisation of parent, child, and reference relation types could be provided with examples (visualised would be much better) to enable the reader to understand definitions and tasks clearly.
>
> We agree with you that examples can help readers understand better. For this aim, we provide detailed categorizations of the _parent_, _child_, _sequence_, and _reference_ relation types (in L232 - L246 of the main paper). For example:
>
> - **Parent Relation**: A section header is the parent of its subsection headers.
> - **Child Relation**: Paragraphs within a section are children of that section header.
> - **Reference Relation**: When text refers to a figure or table, this forms a reference relation.
>
> Apart from the detailed descriptions, a few visualization examples are illustrated in **Figure 3**, including all four relationships, _i.e._, _parent_, _child_, _sequence_, and _reference_ relation.

---

> ### Author Response · Authors · 2024-11-26
> **Response to Reviewer jDZA - Part 2**
>
> **\[W4\] Details of Document Relation Graph Generator (DRGG)**
>
> > The description of Document Relation Graph Generator is not clearly described to me. The overall workflow with the corresponding aim is relatively hard to understand. I have gone through it several times but still hard to understand including input to the framework, why the encoder and encoder structure, and how the decoder works during training and inferencing.
>
> Thank you very much for your feedback. We have carefully revised the explanation of the Document Relation Graph Generator (DRGG) in our revised version. For example, in Figure 5, an input document image will be added at the beginning of the architecture, which is first processed through the **backbone** of the DLA model to extract visual features. Then, these features are passed to the Encoder-Decoder architecture for layout analysis, where the decoder additionally outputs object queries, representing document layout elements. These object queries (orange boxes) are subsequently forwarded to our DRGG. The Relation Feature Extractor (a core DRGG component) processes the object queries via pooling layers and MLPs to generate relational features, capturing both spatial and logical relations. These features are aggregated across decoder layers into a unified representation, which is then passed to the Relation Predictor (another DRGG component) to generate the final relational graph. To further clarify, we will update Figure 5 to clearly label each step of the process.
>
> ---
>
> **\[W5\] Detailed experiment analysis**
>
> > The experimental section is underexplored. Analysis of different document domains, fine-grained relation type analysis (section-&gt;subsection, section-&gt;paragraph).
>
> The detailed results of six different document domains (_i.e._, **Financial Reports**, **Scientific Articles**, **Laws and Regulations**, **Government Tenders**, **Manuals**, and **Patents**) are presented in the tables below.  We used InternImage as the backbone, RoDLA as the detector, and DRGG for relationship extraction. The tables below summarize the performance in terms of mRg and mAPg under relation confidence thresholds of 0.5, 0.75, and 0.95 under the IoU threshold of 0.5.
>
> **mRg Results (Mean Recall for relational graph)**:
>
> | Relation confidence thresholds | **Financial Reports** | **Scientific Articles** | **Laws and Regulations** | **Government Tenders** | **Manuals** | **Patents** |
> | ------------------------------ | --------------------- | ----------------------- | ------------------------ | ---------------------- | ----------- | ----------- |
> | 0.5                            | 15.0                  | 46.3                    | 38.7                     | 40.6                   | 40.6        | 22.7        |
> | 0.75                           | 12.3                  | 42.0                    | 36.5                     | 38.7                   | 35.6        | 20.5        |
> | 0.95                           | 9.0                   | 35.6                    | 33.5                     | 34.1                   | 27.1        | 17.5        |
>
> **mAPg Results (Mean Average Precision for relational graph)**:
>
> | Relation confidence thresholds | **Financial Reports** | **Scientific Articles** | **Laws and Regulations** | **Government Tenders** | **Manuals** | **Patents** |
> | ------------------------------ | --------------------- | ----------------------- | ------------------------ | ---------------------- | ----------- | ----------- |
> | 0.5                            | 52.6                  | 54.5                    | 63.2                     | 55.9                   | 46.8        | 31.8        |
> | 0.75                           | 50.9                  | 52.9                    | 58.7                     | 51.4                   | 44.4        | 30.7        |
> | 0.95                           | 20.2                  | 47.5                    | 54.6                     | 48.1                   | 32.5        | 29.3        |
>
> The results demonstrate clear domain-specific trends. **Laws and Regulations** achieve the highest **mAPg@0.5** with 63.2, benefiting from their structured and consistent layouts, while **Patents** perform worst, with **mRg@0.95** at 17.5, due to their dense and complex layouts. Both **mRg** and **mAPg** decline as the relation confidence threshold increases, reflecting the challenges of capturing precise relationships under stricter criteria. These findings highlight the varying complexities across domains and the need for robustness in handling diverse document structures.
>
> For a detailed fine-grained relation type analysis of the dataset,  please refer to the heatmap in supplementary Subsection **A.2 Detailed Statistics of GraphDoc Dataset.** This analysis illustrates the distribution and frequency of relation types across the dataset, highlighting domain-specific trends.

---

> ### Author Response · Authors · 2024-11-26
> **Response to Reviewer jDZA - Part 3**
>
> **\[W6\] More qualitative analysis**
>
> > Similar to quantitative analysis, more qualitative analyses are expected. I really like the idea of the paper, but more in-depth analysis is essential to maximising the contributions of the author desired.
>
> We thank Reviewer jDzA for the suggestion. There are two visualization examples presented in the Supplementary Material. Please kindly refer to **Section F Qualitative Results of DRGG** and incorporate **Figure 7**. Thanks to your suggestions, we will add more visualization examples to showcase the in-depth analysis of the qualitative results. For example, below are some findings concluded from the analysis of qualitative results from DRGG:
>
> 1. **Capability in relations extraction**: As shown in Figure 7, DRGG effectively captures spatial and logical relationships, _e.g._, parent-child relations between “Picture” and “Caption”, which are essential for reconstructing document structures.
> 2. **Challenges in Dense Layouts**: DRGG faces challenges in densely populated or cluttered layouts, where dense elements hinder its ability to capture all relationships, as seen in cases with multiple section headers, captions, and images.
> 3. **Limitations from DLA**: DRGG's performance is partially constrained by errors from the Document Layout Analysis (DLA) task. As in the second example of Figure 7, the DLA misprediction of two tables as a single table caused missing relations among them.
> 4. **Multimodality improvement**: Incorporating advanced contextual embeddings (_e.g._, text-based features) and enhancing the DLA backbone with better segmentation and classification accuracy can significantly reduce errors and improve relational predictions.
> 5. **Multi-Page extension**: Extending DRGG to support multi-page relational understanding is a key future direction to enhance its applicability and provide a more comprehensive document structure analysis.
>
> ---
>
> **\[Q1\] Gains of spatial relationship rather than the bounding box coordinated**
>
> > Why do you think spatial relation should be considered as one of the relation types of this dataset which might be directly acquired by leveraging the predicted bbox coordinated?
>
> We consider spatial relations essential because spatial relations based on image, _e.g._, derived directly from bounding box coordinates, differ significantly from those based on document paper, which is crucial for understanding and analyzing document structures. For example, in scenarios like _rotated_ or _skewed_ documents, the raw bounding box coordinates may not accurately reflect the true spatial hierarchy or sequence of document layouts. Document-based spatial relations capture the contextual relationships that are fundamental to document understanding, beyond what raw coordinates can provide. Based on this observation, we found that explicitly considering the spatial relation benefits the document structure analysis task.
>
> ---
>
> **\[Q2\] Statistics details of human check**
>
> > Could you provide more statistics details before and after human checking?
>
> As mentioned in W2, we conducted extensive human verification and refinement, covering approximately 58.5% of the dataset. Specifically, we reviewed 4,852 pages of Government Tenders, 12,000 pages of Financial Reports, 6,469 pages of Patents, and 8,000 pages from other domains. The refinement rates for relation labels varied across domains: approximately 23% for Financial Reports, 8% for Scientific Articles, 26% for Government Tenders, and 17% for Patents. Based on our comprehensive cross-validation evaluation, we believe that our dataset is high-quality for the proposed gDSA task. We hope our new dataset and benchmark can provide an innovative advancement in DSA and document an understanding research field.
>
> ---
>
> **\[Q3\] Rules updated in different domain adaptation**
>
> > Do you think the rules might be updated based on different document domains?
>
> Yes, we agree that rules may need to be updated according to different document domains. This is precisely why we incorporate human verification and modification into our process, ensuring that domain-specific nuances are accurately captured and the resulting relationships are reliable. For this aim, we first apply a general rule-based relation extraction system to guarantee the consistency of relation label generation. Then, we verify and refine the automatically generated results to ensure that domain-specific nuances are appropriately addressed across different domains.

---

> ### Author Response · Authors · 2024-11-26
> **Response to Reviewer jDZA - Part 4**
>
> **\[Q4\] Reason for single-page datasets**
>
> > For the relation prediction dataset, is there any reason you choose single-page datasets? It is more natural to use multipage-scenarios to establish the entire document's logical structure.
>
> In this work, we focus on single-page datasets as they provide diverse document structures and layout cases, making them effective for evaluating spatial and logical relationship extraction. As known by us, exploring the graph-based relation of document images is under-explored in the literature. Therefore, we start with the single-page setting. Additionally, a single-page DSA serves as a foundational step, as multipage document structures can be built on top of that by linking relationships across individual pages. In line with your valuable suggestion, we are extending our approaches to multipage gDSA, including the new multipage dataset.

---

> > ### Comment · Reviewer_jDzA · 2024-12-03
> >
> > Thanks for providing a detailed explanation which addressed some of my concerns. However, there are a few logical structures parsing datasets in this domain including the dataset introduced by Docparser and HRDoc. Thus, the contribution of this work is weakened, but I will slightly increase my mark.

---

> > > ### Author Response · Authors · 2024-12-03
> > >
> > > Dear Reviewer jDzA,
> > >
> > > Thank you very much for your feedback, and we are glad that our explanations can address some of your previous concerns. Regarding your concern about the logical structure definitions in datasets like Docparser and HRDoc, we would like to clarify how our new dataset differentiates itself and addresses the limitations of existing datasets.
> > >
> > > (1) While these datasets focus primarily on the relationships between paragraph texts and section headers within the main body of documents, they do not include the relations between document annotation texts and non-textual elements, e.g., *footnotes*, *figures*, and *tables*. In our GraphDoc, we include all text and non-textual document elements.
> > >
> > > (2) These two datasets are limited in considering only parent-child relations, without exploring a broader set of relational expansions within the document. In contrast, our GraphDoc dataset includes eight relationships from both spatial and logical perspectives.
> > >
> > > (3) They mainly focus on hierarchical structures and overlook the coexistence of multiple relations of the same elements from a graph-based perspective. Compared to them, our method adopts a graph-based perspective to model different relations, allowing for multiple relational types. This ensures that our GraphDoc dataset facilitates deeper and more nuanced document analysis.
> > >
> > > These limitations highlight the need for a more comprehensive dataset that captures the complex interplay between various document elements. As a new contribution to this field, we hope that our new dataset can foster the development of document understanding.
> > >
> > > We appreciate your insights and hope the clarification of distinctions and contributions can address your concern. We ensure that all the issues will be clarified in our final version. Thank you for your thoughtful consideration.
> > >
> > > Sincerely,
> > >
> > > Authors

---

### Author Response · Authors · 2024-11-26
**Summary of Revision**

We sincerely thank all reviewers for their valuable feedback and constructive comments. Your suggestions significantly help us to improve the quality of our manuscript. We are a bit late in responding, as we have been actively organizing additional analyses of our previous experiments and conducting new ones.

Thanks to the reviewers' suggestions, we have carefully addressed each concern or question individually and revised our paper accordingly, with **the changed content marked in red** and **the added text marked in orange**. Below, we summarize the key updates:

- In Sec. 3.2 DRGG, we have updated the explanation of the DRGG structure and workflow.
- In Supplementary Sec. A.1, we have added a detailed explanation of the human verification and refinement process.
- In Supplementary Sec. A.2, we have revised the explanation of spatial relations to clearly justify our decision to use four types of spatial relations.
- In Supplementary Sec. C, we have included a detailed figure illustrating the DRGG structure.
- In Supplementary Sec. D, we have added additional experimental results of DRGG across different document domains and relation types.
- In Supplementary Sec. E, we have clarified the analysis section of the previous ablation study and included new ablation study results along with the corresponding analysis.
- In Supplementary Sec. G, we have enhanced and expanded the detailed analysis of the qualitative results of DRGG and provided suggestions for potential improvements.

We hope the revised version can address all concerns of reviewers. Thank you again for your time and consideration.

---

### Meta-Review · Area_Chair_4dFX · 2024-12-20

**Metareview:**

This paper introduces a novel task called Graph-based Document Structure Analysis, which extends traditional document layout analysis by not only detecting document elements but also modeling their spatial and logical relationships as a graph structure. Additionally, the authors present GraphDoc, a comprehensive dataset that includes relational annotations across diverse categories.

The reviewers recognized the significance and originality of this contribution, particularly highlighting the introduction of the proposed task and the extensive dataset. While some concerns were initially raised regarding the clarity of key concepts and the need for more detailed analyses, the authors thoroughly addressed these issues in their revisions. They provided additional explanations, experimental results, and qualitative analyses, which further strengthened the paper. Considering the novelty of the work, its potential impact on the field, and the satisfactory responses to the reviewers' feedback, I recommend this paper for acceptance.

**Additional Comments On Reviewer Discussion:**

During the reviewer discussion, Reviewer jDzA raised concerns about the importance of spatial relations and the limited number of relation types. The authors justified the significance of spatial relations and explained their choice of a concise and generalizable framework.

Reviewer HRLr sought details about the dataset’s human verification process, and the authors highlighted their efforts to ensure high-quality annotations through substantial manual refinement.

Reviewer G241 recommended comparisons with existing models and discussing potential limitations. The authors demonstrated the necessity of their specialized model and addressed scalability and robustness concerns.

In summary, the authors effectively addressed the reviewers’ concerns, enhancing the paper and supporting its acceptance.

---

### Decision · Program_Chairs · 2025-01-22

Accept (Poster)